# Unsupervised Training of Convex Regularizers using Maximum Likelihood Estimation

**Hong Ye Tan**                                                       *hyt35@cam.ac.uk*
*Department of Applied Mathematics and Theoretical Physics*
*University of Cambridge, United Kingdom*

**Ziruo Cai**                                                   *sjtu_caiziruo@sjtu.edu.cn*
*School of Mathematical Sciences*
*Shanghai Jiao Tong University, China*

**Marcelo Pereyra**                                                  *m.pereyra@hw.ac.uk*
*School of Mathematical & Computer Sciences*
*Heriot-Watt University, Edinburgh*

**Subhadip Mukherjee**                                        *smukherjee@ece.iitkgp.ac.in*
*Department of Electronics and Electrical Communication Engineering*
*Indian Institute of Technology, Kharagpur, India*

**Junqi Tang**                                                       *j.tang.2@bham.ac.uk*
*School of Mathematics*
*University of Birmingham, United Kingdom*

**Carola-Bibiane Schönlieb**                                           *cbs31@cam.ac.uk*
*Department of Applied Mathematics and Theoretical Physics*
*University of Cambridge, United Kingdom*

**Reviewed on OpenReview:** *https://openreview.net/forum?id=375tU7TnOS*

## Abstract

Imaging is a canonical inverse problem, where the task of reconstructing a ground truth from a noisy measurement is typically ill-conditioned or ill-posed. Recent state-of-the-art approaches for imaging use deep learning, spearheaded by unrolled and end-to-end models and trained on various image datasets. However, such methods typically require the availability of ground truth data, which may be unavailable or expensive, leading to a fundamental barrier that can not be addressed by choice of architecture. Unsupervised learning presents a powerful alternative paradigm that bypasses this requirement by allowing to learn directly from noisy measurement data without the need for any ground truth. A principled statistical approach to unsupervised learning is to maximize the marginal likelihood of the model parameters with respect to the given noisy measurements. This paper proposes an unsupervised learning approach that leverages maximum marginal likelihood estimation and stochastic approximation computation in order to train a convex neural network-based image regularization term directly on noisy measurements, improving upon previous work in both model expressiveness and dataset size. Experiments demonstrate that the proposed method produces image priors that are comparable in performance to the analogous supervised models for various image corruption operators, maintaining significantly better generalization properties when compared to end-to-end methods. Moreover, we provide a detailed theoretical analysis of the convergence properties of our proposed algorithm.

# 1    Introduction

Image reconstruction often requires solving high-dimensional inverse problems, ranging from image denoising and deconvolution, to phase retrieval or computed tomography. These problems are usually formulated as the recovery of a signal $x \in \mathcal{X}$ from a noisy measurement $y = Ax + w \in \mathcal{Y}$, with forward operator $A : \mathcal{X} \to \mathcal{Y}$ and some noise $w$. A common trait of these problems is ill-posedness, where the solution of $Ax = y$ can have zero or infinitely many solutions, necessitating the use of regularized reconstruction operators (Benning & Burger, 2018; Engl et al., 1996). A classical method of solving this is using variational regularization, combining a data fidelity term with priors such as wavelet priors or total variation priors (Chambolle & Pock, 2016; Arridge et al., 2019). Recently, various end-to-end approaches utilizing neural networks have pushed the state-of-the-art for imaging, achieving highly detailed reconstructions at the cost of well-crafted training data and computational resources. These include direct neural network approaches, as well as iterative plug-and-play priors (Zhang et al., 2021; Venkatakrishnan et al., 2013).

**Variational models.** One way to define the reconstruction of a measurement $y$ is as the solution of a minimization problem $\arg\min_x f_y(Ax) + R_\lambda(x)$ (Arridge et al., 2019; Chambolle & Pock, 2016). Here, $f_y : \mathcal{Y} \to \mathbb{R}$ is a data fidelity function measuring how well the reconstruction $x$ fits the measurement $y \in \mathcal{Y}$, and $R_\lambda : \mathcal{X} \to \mathbb{R}$ is a regularization functional that promotes regularity of the reconstructions. Examples include computed tomography, where $A$ is the Radon transform and $R_\lambda$ may enforce some type of wavelet sparsity (Antoniadis & Fan, 2001; Donoho, 2006). In the particular case of Gaussian denoising, $A = I$, the fidelity can be taken to be $f_y(x) = \|x - y\|^2/2$ and regularization $R_\lambda(x) = \lambda\|\nabla x\|_{1,\mathcal{X}}$, commonly known as the total variation (TV) regularized solution. Gaussian noise allows for another probabilistic interpretation using Tweedie's identity, which provides a relationship between the the posterior mean estimator and a ground-truth density $p(x)$: it states that if $p_\epsilon$ is given by convolving $p$ with a Gaussian kernel of variance $\epsilon$, then the minimum mean-squared estimator (MMSE) denoiser $D_\epsilon^*$ satisfies $(D_\epsilon^* - I)(x) = \epsilon \nabla \log p_\epsilon^*(x)$ (Efron, 2011; Laumont et al., 2022). That is, the denoising step is in the direction of the posterior score, precisely given by gradient descent on a convolved version of $f_y(x) + g(x)$. This begins to endow learnable variational models with probabilistic interpretation.

Learned variational models are a middle ground between fully classical model-based reconstructions and fully learned end-to-end models (Lunz et al., 2018; Mukherjee et al., 2020; Kobler et al., 2020). Heuristically, the fidelity term $f_y(Ax)$ should depend on the observation process involving forward operator $A$ and the noise model, while the regularization $R_\lambda$ ignores the observation process and considers only the data. One advantage of the decoupling is that the learned regularizer can be applied to different forward models without retraining. This is exploited in Plug-and-Play methods, which implicitly use generic denoisers as regularizers for a variety of non-denoising tasks such as demosaicing and super-resolution (Zhang et al., 2021; Hurault et al., 2022a). Moreover, the variational structure allows for the application of classical regularization theory under regularity assumptions on $R_\lambda$ (Burger & Osher, 2004; Hauptmann et al., 2024; Schwab et al., 2019).

**Learned regularization.** Notable examples of learned regularization include adversarial regularization and network Tikhonov (Mukherjee et al., 2021; Lunz et al., 2018; Li et al., 2020a; Alberti et al., 2021). In the supervised regime, i.e. presence of clean and noisy training pairs, one method is to minimize some distance metric between the clean image and the reconstructed noisy image as given by an approximate solution of the variational problem (Diamond et al., 2017). Additional regularization is often added to enforce regularity on the learned regularizer, such as having small Lipschitz constant, which is linked to Wasserstein robustness (Kuhn et al., 2019). In the regime of weak supervision, given unpaired clean and noisy data, adversarial regularization aims to penalize noisy images and promote clean images by maximizing and minimizing the regularizing terms respectively (Lunz et al., 2018; Mukherjee et al., 2020). Using the structure of a fidelity and regularizer allows for a more explicit description of the final reconstruction, and can improve the performance of a learned denoiser when used in an unrolled scheme (Zhang et al., 2021). We note that it is possible to indirectly learn $R_\lambda$ by instead learning mappings such as the proximal $\text{prox}_{R_\lambda}$ or gradient $\nabla R_\lambda$, while still maintaining well-posedness of the variational formulation through convex analysis identities (Gribonval & Nikolova, 2020; Moreau, 1965). For example, under certain Lipschitz conditions, denoisers can be interpreted as proximal operators of weakly convex functions, allowing for another degree of regularity (Hurault et al.,

2022b; Shumaylov et al., 2023). This denoiser-prior connection has been recently extended to diffusion models (Feng et al., 2023; Graikos et al., 2022).

**Bayesian approaches.** Approaches rooted in the Bayesian statistical paradigm usually aim to characterize the posterior distribution (density) of $x$ conditioned on observing the noisy measurement $y$ (Stuart, 2010; Calvetti & Somersalo, 2018). Using Bayes' rule, this density can be expressed as $p(x|y) = \ell(y|x)p(x)/\mathrm{Z}(y)$, where $\ell(y|x) = p_w(y, Ax)$ is the likelihood arising from the noise distribution of $w$, $p(x)$ is the prior density on $x$, and $\mathrm{Z}(y)$ is a normalizing constant. The modelled distribution is thus characterized by a choice of family for the prior $p(x)$. Computational Bayesian inference has been an active area of development for the past 40 years, including hierarchical image models (Geman & Geman, 1986; Carlin et al., 1992), diffusions (Roberts & Stramer, 2001), and dynamical systems (Friston, 2002; Wilkinson, 2018), and we refer to Green et al. (2015); Mukherjee et al. (2023) for more comprehensive overviews. A widely-used modern method to describe the distribution $p(x|y)$ is to draw samples from it using Markov Chain Monte Carlo (MCMC) methods (Gilks et al., 1995; Robert et al., 1999; 2007). One distinct advantage of the Bayesian approach is the option of uncertainty quantification rather than point estimation, given sufficient computational resources (Carioni et al., 2023; Nagel & Sudret, 2016; Durmus et al., 2018; Holden et al., 2022). This is used in various imaging methods such as Wasserstein generative adversarial networks (Arjovsky et al., 2017), and Cycle-GANs (Zhu et al., 2017), which utilize optimal transport to flow from noisy to clean images with a learned prior. Since Bayesian approaches use the entire distribution, another possible advantage over other training methods is that paired training examples (such as clean-noisy pairs) are often not needed, requiring only access to samples from the marginal distributions and not the joint distribution. This can be useful in tasks such as unsupervised image registration, where paired data is labor-intensive (Balakrishnan et al., 2018; Chen et al., 2022b).

The variational formulation can be interpreted as a special case of the Bayesian formulation. Given the negative prior log-density $g(x) = -\log p(x)$ and negative log-likelihood $f_y(x) = -\log \ell(y|x)$, the maximum a-posteriori estimator (MAP) $x_{\mathrm{MAP}} = \arg\max_x p(x|y) \propto \exp(-f_y(x) - g(x))$ is equal to the minimizer $\min_x f_y(x) + g(x)$. In Bayes' rule, the likelihood can thus be interpreted as a fidelity term, and the prior as regularization. Pereyra (2019) further explores this connection, demonstrating that for log-concave distributions, the MAP estimator is optimal with respect to a Bregman divergence in terms of expectation, and is the dual problem of an MMSE problem. This connects variational regularization with the statistical problem of MAP estimation.

**Unsupervised learning.** Many end-to-end approaches as well as approaches described above require clean ground truths $x$ and measurements $y$, which can be limiting in cases where ground truth data is unavailable or expensive, such as in medical imaging. We are instead concerned with the situation where we have a finite amount of noisy data, such as if a clean dataset is corrupted. One such approach is the deep image prior (DIP), in which the a neural network is fed a random input, and subsequently has its parameters optimized to recover a noisy measurement with early stopping (Ulyanov et al., 2018). This has shown remarkable performance without any supervised data, using only the implicit regularization of the neural network architecture. Theoretical results for DIP include optimization dynamics analysis (Heckel & Soltanolkotabi, 2020; Van Veen et al., 2018) and Bayesian interpretations using Gaussian process limits (Cheng et al., 2019). Pajot et al. (2018) propose using the discriminator of a GAN as a regularizer, using Stein's unbiased risk estimator to train using only corrupted measurements (Metzler et al., 2018; Stein, 1981). Noise2Inverse considers the specific case where one noisy measurement can be considered as the average of multiple noisier measurements to artificially generate more data, such as in limited-angle tomography (Hendriksen et al., 2020). We note that some unsupervised approaches such as Noise2Noise generate an unlimited number of noisy measurements, where averaging out the empirical risks with the noisy targets leads to an approximation of the empirical risks when training with clean targets (Lehtinen et al., 2018), but we do not consider this simpler setting.

Equivariant imaging (EI) is another unsupervised framework that learns reconstruction functions from only compressed measurements (Chen et al., 2021; Pereyra & Tachella, 2024; Scanvic et al., 2023). The framework utilizes the fact that the set of plausible signals in $\mathcal{X}$ are invariant to a certain group of transformations $\mathcal{G}$. By "rotating" the range of the adjoint operator, or constructing virtual operators corresponding to invariances of the forward operator, the learned operator is able to learn reconstructions that are approximately equivariant

to the transformations. This greatly mitigates problems arising from nontrivial nullspaces and removes artifacts that are not invariant under transformations in $\mathcal{G}$.

We briefly remark on unsupervised versions of diffusion models, which is an area of intense research efforts. Diffusion models trained in a supervised manner have been successfully deployed in a plug-and-play manner with a data guidance term, delivering state-of-the-art reconstructions across a variety of challenging imaging problems (Chung et al., 2023; Batzolis et al., 2021; Mardani et al., 2024). However, from a Bayesian computation viewpoint, these "guided" diffusion models are highly inaccurate (e.g., they significantly underestimate uncertainty quantification results, see (Pereyra & Tachella, 2024; Thong et al., 2024)). Alternatively, for some problems it is also possible to employ Stein's unbiased risk estimator to train diffusion models directly from measurement data, without ground truth (Kawar et al., 2024; Aali et al., 2023). However, in problems that are ill-posed, issues arise with the null-space of the forward operator, leading to improper training (Chen et al., 2022a). Other approaches consider transferring powerful pre-trained diffusion models from one setting to another, such as change of noise, forward operator, or dataset (Wang et al., 2023; Zhu et al., 2023). As discussed previously, we are instead interested in the case where we do not have access to multiple copies of corrupted measurements nor a pre-trained model. In this sense, our proposed method is thus able to train priors in strictly harder scenarios.

**Statistical estimation of parameters.** The problem of learning model parameters sits naturally within the statistical estimation framework and can be tackled by using powerful statistical inference techniques, such as maximum likelihood estimation. In many cases, such techniques do not require ground truth data and they are known to outperform common heuristics such as the L-curve criterion (Hansen & O'Leary, 1993) and Morozov's discrepancy principle (Morozov, 1966) (e.g., see Vidal et al. (2020) for a detailed comparison of maximum likelihood estimation and alternative strategies in the context of regularization parameters).

In addition, Bayesian statistical approaches allow for statistical interpretations of reconstructions and their associated parameters. Alberti et al. (2021) provides a statistically motivated method of learning a Tikhonov preconditioner based on covariance estimators. Expectation maximization allows for finding locally optimal parameters when there are unobserved latent variables in an iterative manner (Dempster et al., 1977; Robert et al., 1999), which can be applied to tasks such as image segmentation and tomography (Carson et al., 2002; Levitan & Herman, 1987). Vidal et al. (2020) proposes a stochastic approximation method of estimating the optimal regularisation parameters for TV regularized denoising or hyperspectral unmixing for a single image using maximum marginal likelihood estimation, which can then be used for post-hoc reconstruction. The method of Vidal et al. (2020) was recently extended by Mbakam et al. (2024) to include the estimation of unknown parameters in the forward model for semi-blind image deconvolution problems, and to calibrate deep generative priors within a Bayesian image estimation framework in Melidonis et al. (2024). This paper builds on this to develop a stochastic approximation scheme to train a convex neural network regularization terms in a fully unsupervised manner.

## 1.1 Contributions

In this work, we propose a principled unsupervised method of training convex regularizers with the following objectives.

1. (*Unsupervised.*) Training a regularizing prior using *only noisy measurements*. This is achieved using maximum marginal likelihood estimation. In particular, we do not assume the existence of multiple noisy copies of the data, such as in Noise2Noise schemes (Lehtinen et al., 2018).

2. (*Expressive.*) We increase the dimensionality of the regularizer from the order of $10^1$ to $10^5$. Existing MLE-based methods consider reconstruction for single images by tuning scalar parameters for hand-crafted priors. By using a more expressive parameterization for the regularizer, we can achieve better performance, and demonstrate that the proposed MCMC-based method is applicable in high dimensions.

3. (*Efficient.*) Unsupervised methods have strictly less available information than supervised methods. By comparing the proposed unsupervised method with the corresponding supervised method with the

same regularizer architecture, we demonstrate that this loss of information is almost mitigated. We observe that the regularizers trained in the proposed unsupervised manner have a strong regularizing effect, and have reasonable performance gaps compared to the same model under supervised training.

4. (*Convergent training.*) Theoretical ergodic convergence of the regularizers under the proposed method. This is done using a particular choice of regularizer architecture, which is used to verify the assumptions of convergence results in previous works (De Bortoli et al., 2021; 2020).

5. (*Convergent estimation.*) A convex architecture also implies nice properties of the statistical estimation. In the context of sampling, a convex regularizer leads to a log-concave density and provides existence of all posterior moments, which implies that the estimation of desired statistical quantities is well-posed.

This work is organized as follows. Section 2 casts the variational regularization formulation into the Bayesian framework, and motivates estimating regularization parameters using maximum marginal likelihood estimation. This problem is then decomposed and solved using a stochastic gradient scheme. The proposed method is then given in Algorithm 1, with its mini-batched version Algorithm 2 for use in larger datasets. This extends the work of Vidal et al. (2020), from training regularization parameters on single images, to training convex ridge regularizers on entire datasets in an unsupervised manner. Section 3 presents convergence theory for the proposed method using the convex ridge regularizer architecture. In particular, Theorem 2 states that the proposed stochastic gradient scheme has ergodic convergence. Section 4 evaluates the performance of our proposed method against various supervised and unsupervised baseline methods.

## 2 Unsupervised Learning by Maximum Marginal Likelihood Estimation

In this section, we first reformulate the variational regularization framework as a maximum a-posteriori problem. From this Bayesian interpretation, we consider finding the regularizer parameters using maximum marginal likelihood estimation, which uses only noisy measurements. Motivated by applying gradient ascent on the marginal likelihood, we then introduce the decomposition of the marginal likelihood into posterior and prior expectations as given in Vidal et al. (2020).

Given a fidelity term $f_y$ and a regularization term $g_\theta$ with trained parameters $\theta$, the standard variational reconstruction for a given measurement $y$ is given as follows:

$$\hat{x} = \arg\min_x \left[ f_y(x) + g_\theta(x) \right].$$

We can reformulate this as a maximum a-posteriori problem as mentioned in the introduction. Setting $f_y(x) = -\log \ell(y|x)$ as the negative log-likelihood and $g_\theta(x) = -\log p(x|\theta)$, the variational reconstruction is equivalent to

$$x_{\mathrm{MAP}} = \arg\max_x \log \left[ \ell(y|x) p(x|\theta) \right] = \arg\max_x \log p(x|y, \theta), \tag{1}$$

where we use Bayes' rule: $p(x|y, \theta) \propto \ell(y|x) p(x|\theta)$. To apply this relation for reconstruction, we still need to learn $\theta$. In the unsupervised paradigm, samples from the joint distribution $p(x, y)$ and even the prior distribution $p(x)$ may be unavailable, which leaves us with learning an estimator $\theta^*$ directly from noisy measurements sampled from $p(y)$.

Adopting an empirical Bayesian approach, the regularization parameter $\theta^*$ can be accurately estimated from noisy measurements $y$ by maximum marginal likelihood estimation (Vidal et al., 2020):

$$\theta^* = \arg\max_{\theta \in \Theta} \log p(y|\theta). \tag{2}$$

Here, $p(y|\theta)$ is the marginal likelihood given by marginalizing over data $p(y|\theta) = \int \ell(y|x) p(x|\theta) \mathrm{d}x$, and $p(x|\theta) \propto \exp(-g_\theta(x))$ is the density of the prior Gibbs measure. Standard assumptions include the negative log-likelihood $f_y(x) = -\log \ell(y|x)$ being convex and $\mathcal{C}^1$ with $L_y$-Lipschitz gradient (in $x$), as well as the admissible parameter set $\Theta$ being compact.

One method of maximizing the marginal likelihood is by gradient ascent on $\theta$, which requires knowledge of $\nabla_\theta \log p(y|\theta)$. However, this is intractable and requires approximation. We now detail how to approximately

compute this gradient using only noisy measurements $y$ using a double MCMC approach (Vidal et al., 2020), extending the aforementioned work from single measurements to datasets. The stochastic gradient approximator can then be used inside a gradient-step optimizer, referred to in the literature as the stochastic optimization with unadjusted Langevin (SOUL) algorithm (De Bortoli et al., 2021; 2020).

We first decompose the marginal likelihood $p(y|\theta)$ as follows (Vidal et al., 2020). Denote the normalizing constant of the prior Gibbs measure $p(x|\theta) \propto \exp(-g_\theta(x))$ as

$$Z(\theta) = \int \exp(-g_\theta(x)) \, \mathrm{d}x. \tag{3}$$

Then $p(x|\theta) = \exp(-g_\theta(x))/Z(\theta)$ is the prior density. While the marginal likelihood $\theta \mapsto p(y|\theta)$ is intractable, under certain regularity assumptions[1], detailed in Proposition 1 and Vidal et al. (2020, Prop. A1), it can be replaced with a noisy estimate and decomposed in the following manner using Fisher's identity (Vidal et al., 2020; Douc et al., 2014):

$$\nabla_\theta \log p(y|\theta) = \int p(x|y,\theta) \nabla_\theta \log p(x,y|\theta) \, \mathrm{d}x$$

$$= -\int \nabla_\theta g_\theta(x) p(x|y,\theta) \, \mathrm{d}x - \nabla_\theta \log(Z(\theta)). \tag{4}$$

Noticing that $\nabla_\theta Z(\theta) = -\int \nabla_\theta g_\theta(x) \exp(-g_\theta(x)) \, \mathrm{d}x$, we can write

$$\nabla_\theta \log Z(\theta) = (\nabla_\theta Z(\theta))/Z(\theta) = -\int \nabla_\theta g_\theta(x) p(x|\theta) \, \mathrm{d}x. \tag{5}$$

Putting (4) and (5) together, we have the following integral expression for the marginal likelihood:

$$\nabla_\theta \log p(y|\theta) = \mathbb{E}_{x|\theta}[\nabla_\theta g_\theta(x)] - \mathbb{E}_{x|y,\theta}[\nabla_\theta g_\theta(x)]. \tag{6}$$

Using this decomposition, we are able to use MCMC methods to estimate the prior and normalizing constant, allowing us perform a noisy gradient ascent on $\theta$ using only noisy observations $y$ (and access to the likelihood $\ell$). In this work, we extend the setup of Vidal et al. 2020 in the following manner:

1. Experimentally, the prior distributions are loosened from being of the form $p(x|\theta) \propto \exp(-\theta^\top g(x))$ where $g_\theta = \theta^\top g(x)$ is homogeneous, to the more general parameterized form $p(x|\theta) \propto \exp(-g_\theta(x))$ for some convex $g_\theta$.

2. The dimensions of the parameter space $\Theta$ are increased from $d_\Theta \leq 10$ to the order of $10^5$, the dimensionality of a small convex regularizer.

## 2.1 Jointly Sampling Prior and Posterior

To compute the gradient of the marginal likelihood, we need to evaluate the expectations in (6). In general, this is analytically intractable and needs to be done numerically, though properties such as positive homogeneity can make this easier (Vidal et al., 2020, Sec. 3.3).

First assume that both the likelihood $\ell(y|x)$ and regularizer $g_\theta$ are $\mathcal{C}^1$ (in $x$). Observe that (6) is a difference of expectations of $\nabla_\theta g_\theta(x)$ over two probability distributions, namely over the prior $p(x|\theta)$ and posterior $p(x|y,\theta)$. Following the setup of Vidal et al. (2020), we consider two Markov chains based on the unadjusted Langevin algorithm (ULA) to approximate these expectations (Dalalyan, 2017; Durmus & Moulines, 2017). For step-size parameters $\gamma, \gamma' > 0$, the chains are given as follows, where $Z_k, Z'_k$ are i.i.d. standard Gaussians with identity covariance matrix:

$$R_{\gamma,\theta}: \ X_{k+1} = X_k - \gamma \nabla_x (f_y + g_\theta)(X_k) + \sqrt{2\gamma} Z_{k+1}; \tag{7a}$$

$$\bar{R}_{\gamma',\theta}: \ \bar{X}_{k+1} = \bar{X}_k - \gamma' \nabla_x g_\theta(\bar{X}_k) + \sqrt{2\gamma'} Z'_{k+1}. \tag{7b}$$

---

[1]It is sufficient that $p(y,x|\theta)$ and $\ell(y|x)$ are positive p.d.f.s, $\theta \mapsto p(y,x|\theta)$ is differentiable, and that for any $\tilde{\theta} \in \mathrm{int}(\Theta)$, there is a neighborhood around $\tilde{\theta}$ such that $\|\nabla_\theta p(y,x|\theta)\|$ is uniformly majorized by some $\tilde{g}(x)$ that is integrable w.r.t. $\ell(y|x)$.

The above Markov kernels $R_{\gamma,\theta}$ and $\bar{R}_{\gamma',\theta}$ target the posterior $p(x|y,\theta) \propto \exp(-f_y(x) - g_\theta(x))$ and the prior $p(x|\theta) \propto \exp(-g_\theta(x))$ respectively. This allows us to use Monte Carlo integration in (6), and approximate the marginal posterior gradient $\nabla_\theta \log p(y|\theta)$. We note that it is possible to lift the smoothness requirement on $\ell$ or $g_\theta$ by using a different Markov kernel such as the Moreau–Yosida unadjusted Langevin algorithm (MYULA, Durmus et al., 2018) or the proximal unadjusted Langevin algorithm (PULA, Durmus et al., 2019). With the approximate marginal posterior gradient, we can apply stochastic gradient descent on $\theta$ with step-sizes $\delta_n$, yielding the stochastic approximation proximal gradient (SAPG–ULA) Algorithm 1, as presented in Vidal et al. (2020).

---

**Algorithm 1** SAPG–ULA

---

**Require:** Initial $\{\theta_0, X_0^0, \bar{X}_0^0\}$, $(\delta_n, m_n)_{n \in \mathbb{N}}$, $\Theta$, MC step-size parameters $\gamma, \gamma'$, iterations $N$
1: **for** $n = 0$ to $N - 1$ **do**
2:     **if** $n > 0$ **then**
3:         Set $X_0^n = X_{m_{n-1}-1}^{n-1}$                                              ▷ Posterior chain
4:         Set $\bar{X}_0^n = \bar{X}_{m_{n-1}-1}^{n-1}$                                            ▷ Prior chain
5:     **end if**
6:     **for** $k = 0$ to $m_n - 1$ **do**
7:         Sample $X_{k+1}^n \sim R_{\gamma,\theta_n}(X_k^n, \cdot)$                              ▷ Update posterior chain
8:         Sample $\bar{X}_{k+1}^n \sim \bar{R}_{\gamma',\theta_n}(\bar{X}_k^n, \cdot)$                      ▷ Update prior chain
9:     **end for**
10:    Set $\theta_{n+1} = \Pi_\Theta[\theta_n + \frac{\delta_{n+1}}{m_n} \sum_{k=1}^{m_n} \{\nabla_\theta g_\theta(\bar{X}_k^n) - \nabla_\theta g_\theta(X_k^n)\}]$          ▷ SOUL step
11: **end for**

---

In Step 10, the estimated parameter $\theta_n$ is updated with gradient ascent on $p(y|\theta)$, where the expectations in (6) are approximated with Monte Carlo integration with $m_n$ samples, done in Steps 6 to 9. A projection $\Pi_\Theta$ onto the compact set $\Theta$ is then imposed after each update of $\theta$. The following result states that convergence is attained when the Markov chain updates and the updates for $g_\theta$ happen in an alternating fashion. Moreover, a single sample for the Monte Carlo integration over the prior and posterior is sufficient for convergence. This is key to computational efficiency, by reducing the number of samples needed from the Markov chain.

**Theorem 1** (De Bortoli et al. 2020, Theorem 6). *Suppose that $-\log p(y|\theta)$ is convex w.r.t. $\theta$, and that Assumptions 1 to 3 hold. Under certain technical Lipschitz conditions and decaying step-sizes, a single sample is sufficient, i.e., $m_n = 1$ leads to almost sure convergence of $(\theta_n)_{n \in \mathbb{N}}$ to some maximizer $\theta_* \in \arg\max_\Theta p(y|\theta)$.*

We restate another version of this theorem in a more formal manner in Section 3. In particular, we verify that the required assumptions hold when using the convex ridge regularizer architecture for $g_\theta$. This is done in the constant step-size setting, which instead gives convergence in expectation.

## 2.2 Reflecting Markov Chains

While the Markov chains (7) sample from biased versions of the target posterior and prior distributions in the case of unrestricted domains, this does not translate to constrained sampling. Indeed, the natural domain for images requires that pixel values be non-negative. Negative pixels may cause problems in particular for the case of Poisson imaging, where the likelihood is zero for negative measurements. Melidonis et al. (2022) consider modifying the chains to force the Markov chain samples to adhere to the non-negativity constraint by projecting or reflecting into the target domain. A minor modification of Equation (7) leads to the following reflected Markov chains, where the absolute value of a vector $|v|$ is to be taken componentwise:

$$R_{\gamma,\theta}: \ X_{k+1} = |X_k - \gamma\nabla_x(f_y + g_\theta)(X_k) + \sqrt{2\gamma}Z_{k+1}|; \tag{8a}$$

$$\bar{R}_{\gamma',\theta}: \ \bar{X}_{k+1} = |\bar{X}_k - \gamma'\nabla_x g_\theta(\bar{X}_k) + \sqrt{2\gamma'}Z'_{k+1}|. \tag{8b}$$

We can rewrite the target densities in the following manner. Denote $\iota_+(x)$ as the characteristic function of the positive orthant $C = \mathbb{R}^d_+$:

$$\iota_C(x) = \begin{cases} 0, & x \in C, \\ +\infty, & x \notin C. \end{cases}, \quad \iota_+ \coloneqq \iota_{\mathbb{R}^d_+}, \quad \mathbb{R}^d_+ = \left\{ x \in \mathbb{R}^d \mid x_i > 0, \ i = 1, ..., d \right\}. \tag{9}$$

Let $E_\theta(x) = f_y(x) + g_\theta(x)$. From the above reflections on the boundary of $\mathbb{R}^d_+$, the original target distribution $p(x|\theta)$ and $p(x|y,\theta)$ are truncated and modified into the following densities:

$$p(x|y,\theta) = \frac{e^{-(E_\theta(x)+\iota_+(x))}}{\int_{\mathbb{R}^d_+} e^{-E_\theta(\tilde{x})} \mathrm{d}\tilde{x}}, \quad p(x|\theta) = \frac{e^{-(g_\theta(x)+\iota_+(x))}}{\int_{\mathbb{R}^d_+} e^{-g_\theta(\tilde{x})} \mathrm{d}\tilde{x}}. \tag{10}$$

It can be shown that under suitable assumptions, the reflected SDE admits a unique invariant measure on $\mathbb{R}^d_+$, and the invariant density is given by (10) (Melidonis et al., 2022, Thm 3.4). The Markov chains (8a) and (8b) are the discrete counterparts of the reflected SDE on $\mathbb{R}^d_+$ constructed in (Melidonis et al., 2022).

We note that reflection onto the positive orthant is preferred over projection since we work with densities. In particular, projection will assign positive probability to the boundaries of the target domain $\mathbb{R}^d_+$, which has zero Lebesgue measure. Therefore, the law of the projected SDE will not admit a density with respect to the Lebesgue measure, invalidating the requirements of presented theory. We leave the analysis of the reflected Markov chains to future work.

## 2.3 Extensions to Datasets – Mini-Batching

SAPG–ULA, Algorithm 1, is the basic method for solving the MLE problem (2) in the case where $g_\theta$ needs only be trained on single datapoints, where the problem is sufficiently well-posed. The Markov kernels can be chosen to be either standard ULA (7) or reflected ULA (8) as applied in Vidal et al. (2020); Melidonis et al. (2022) respectively. However, these implementations are limited to reconstructions for single images, as well as only one or two scalar parameters for $\theta$. In the case where $g_\theta$ is a neural network, the MLE problem for a single image may be severely overparameterized, leading to slow convergence. Inspired by traditional machine learning, we aim to train a neural network regularizing prior with more data, such as on a standard image dataset like STL-10.

The first computational difficulty comes from Step 10 of Algorithm 1, where $\nabla_\theta g_\theta$ is needed. Since computing gradients of $g_\theta$ over all images in the dataset is computationally infeasible, mini-batching is necessary in practice, similarly to how stochastic methods such as SGD or Adam are applied rather than full-batch gradient descent. In this section, we present a straightforward extension of the SAPG scheme to the batched case in Algorithm 2, by using multiple posterior Markov chains and changing the order in which the prior Markov chains are updated. This has the upside of only having to store one mini-batch worth of gradients at any given time, which alleviates memory issues by updating $g_\theta$ in a stochastic manner.

Suppose that we are given fixed batched noisy measurements $\{Y_b\}_{b=1}^B$, such as when a noisy image dataset is partitioned. The batched SAPG scheme consists of multiple posterior Markov chains $\{X^{n,b}\}_{b=1}^B$, where the posterior chains $X^{n,b}$ are updated according to $R_{\gamma,\theta}$. Only one prior chain $\bar{X}^n$ is employed, updated according to $\bar{R}_{\gamma,\theta}$. One chain is sufficient since the prior Markov kernel $\bar{R}$ does not depend on the measured data $Y_b$, saving memory compared to running multiple prior Markov chains.

The proposed batched SAPG–ULA method is summarized in Algorithm 2. This extends the SAPG–ULA scheme from single images to datasets. The Markov kernels $R_{\gamma,\theta_n}, \bar{R}_{\gamma',\theta_n}$ need only have stationary distribution approximating the target posterior $p(x|y,\theta)$ and prior $p(x|\theta)$ respectively, which can be of the form (7), (8), or another kernel which can be chosen based on the regularity properties of the target log-likelihood and prior.

Compared to classical supervised training, we make a few approximations in Algorithm 2. The first approximation comes from the lack of ground-truth data, which is replaced with the prior given by the regularizer $g_\theta$. Another approximation comes from approximating the expectation over prior/posterior with only a single sample, motivated by Theorem 1. There will also be bias induced by the choice of ULA for Markov chain, as well as the joint updates of parameters $\theta$ and Markov chains.

---

**Algorithm 2** Batched SAPG–ULA

---

**Require:** Number of batches $B$, batched measurements $\{Y_b\}_{b=1}^B$, Initial $\{\theta_0, X_{0,b}^0, \bar{X}_0^0\}_{b=1}^B$, $(\delta_n, m_n)_{n \in \mathbb{N}}$, $\Theta$, parameters $\gamma, \gamma'$, iterations $N$,

1: **for** $n = 0$ to $N - 1$ **do**
2:     **if** $n > 0$ **then**
3:         Set $X_0^{n,b} = X_{m_n-1}^{n-1,b}$ for $b \in [B]$               $\triangleright$ Posterior chain for each batch
4:         Set $\bar{X}_0^n = \bar{X}_{m_n-1}^{n-1}$                             $\triangleright$ Single prior chain
5:     **end if**
6:     **for** $k = 0$ to $m_n - 1$ **do**
7:         Sample $\bar{X}_{k+1}^n \sim \bar{R}_{\gamma',\theta_n}(\bar{X}_k^n, \cdot)$                $\triangleright$ Update prior chain
8:         **for** $b = 1$ to $B$ **do**
9:             Sample $X_{k+1}^{n,b} \sim R_{\gamma,\theta_n}(X_k^{n,b}, \cdot; Y_b)$       $\triangleright$ Update all posterior chains
10:         **end for**
11:     **end for**
12:     $\theta_{n+1} = \Pi_\Theta[\theta_n + \frac{\delta_{n+1}}{m_n} \sum_{b=1}^B \sum_{k=1}^{m_n} \{\nabla_\theta g_\theta(\bar{X}_k^n) - \nabla_\theta g_\theta(X_k^{n,b})\}].$     $\triangleright$ Accumulated-grad SOUL step
13: **end for**

---

### 2.4 Connection with Adversarial Regularization

We additionally note a connection with adversarial regularization. Under integrability conditions and by Fubini's theorem, (6) shows that the problem of maximizing the marginal likelihood $p(y|\theta)$ in (2) is equivalent to maximizing

$$\sup_\theta \mathbb{E}_{x|\theta}[g_\theta(x)] - \mathbb{E}_{x|y,\theta}[g_\theta(x)]. \tag{11}$$

The Bayesian MLE problem (11) can be thought of as maximizing the value of the regularizer weighted over all possible $x$ (given by the prior expectation), while minimizing the regularizer for good reconstructions (given by the posterior expectation). This should be contrasted with adversarial regularization (Lunz et al., 2018), which has loss functions of the form

$$\sup_\theta \mathbb{E}_{x \sim \text{noisy}}[g_\theta(x)] - \mathbb{E}_{x \sim \text{true}}[g_\theta(x)]. \tag{12}$$

Comparing, the expectations over the noisy and ground-truth distributions of the adversarial loss (12) are replaced with versions that depend on the parameter $\theta$, in particular, the prior and posterior distributions in (11). The prior can be interpreted as the family of all noisy reconstructions, and the posterior as reconstructions given measurements, which replaces ground-truth images. The Bayesian MLE problem can thus be formulated as adversarial regularization between the prior and posterior distributions.

The difference of expectations form of (11) and (12) are also related to the Wasserstein-1 metric, which is a notion of distance between two finite measures. This connection is noted for example in Arjovsky et al. (2017). For two probability measures on $\mathbb{R}^d$, the definition of the Wasserstein-1 metric $\mathcal{W}_1$ and Kantorovich-Rubenstein duality result are follows (Ambrosio et al., 2013; Villani, 2009):

$$\mathcal{W}_1(\mu, \nu) = \inf_{\gamma \in \Gamma(\mu,\nu)} \int \|x - y\| \, \mathrm{d}\gamma(x, y) = \sup_{g \in 1-\text{Lip}} \left[ \int g \, \mathrm{d}\mu - \int g \, \mathrm{d}\nu \right], \tag{13}$$

where $\Gamma(\mu, \nu)$ is the set of all couplings, i.e. probability measures on $\mathbb{R}^d \times \mathbb{R}^d$ with marginals $\mu$ and $\nu$ respectively. Therefore, if the suprema in (11) and (12) were instead over $g_\theta$ that are 1-Lipschitz, the problems are reduced to finding the so-called Kantorovich potentials that achieve the supremum in (13) between the prior and posterior distributions, or the noisy and true distributions respectively.

## 3 Convergence Analysis

In this section, we first introduce the architecture used for $g_\theta$. We then state a summary of the main theorems of De Bortoli et al. (2021), which demonstrates convergence in expectation to the minimizer, with

the associated assumptions provided afterwards. The required assumptions will then be verified for the convex ridge regularizer, which together show convergence of SAPG for our formulation. We provide short intuitive explanations for each assumption and how the regularizer satisfies them, with detailed proofs left to Appendix A.

The regularizer architecture used for these experiments is the convex ridge regularizer (CRR) (Goujon et al., 2022). This takes the form of a one-layer network:

$$g_\theta : \mathbf{x} \mapsto \sum_{i=1}^{C} \psi_i(\mathbf{w}_i^\top \mathbf{x}), \tag{14}$$

where with some abuse of notation, $\psi_i : \mathbb{R} \to \mathbb{R}$ are convex profile functions (ridges) applied element-wise, and $\mathbf{w}_i \in \mathbb{R}^d$ are learnable weights. The parameters $\theta$ consist of the weights $\mathbf{w}_i$ and parameters of the ridges $\psi_i$. The gradient of the regularizer can be expressed as

$$\nabla g_\theta(\mathbf{x}) = \mathbf{W}^\top \sigma(\mathbf{W}\mathbf{x}), \tag{15}$$

where $\mathbf{W} = [\mathbf{w}_1 \ldots \mathbf{w}_C]^\top \in \mathbb{R}^{C \times d}$ is linear, and $\sigma : \mathbb{R}^C \to \mathbb{R}^C$ is a component-wise activation function with components $\sigma_i = \psi_i' : \mathbb{R} \to \mathbb{R}$. Note that due to the convexity of $\psi_i$, the derivatives $\sigma_i$ are increasing. By restricting the parameters such that $\sigma_i$ are strictly increasing, the CRR can be made to be uniformly strongly convex.

For ease of notation, first suppose we have sufficient regularity such that Fisher's identity (11) holds, which will be justified in Proposition 1 and Assumption 3. Let the negative log joint density $-\log p(x, y|\theta)$ be given by

$$E_\theta(x) := f_y(x) + g_\theta(x). \tag{16}$$

Note that $\nabla_x E_\theta(x)$ is the diffusion term of the Langevin step (7a) from posterior $p(x|y, \theta)$, while $\nabla_x g_\theta(x)$ is the diffusion term of the Langevin step (7b) from prior $p(x|\theta)$. We will use ergodicity of both chains to demonstrate (biased) convergence in expectation of $\theta$.

In the following theorem, we consider the scenario where a fixed $m_n = 1$ is used in Algorithm 1 for the Monte Carlo integration, and a fixed step-size $\gamma > 0$ is used for the Markov chains. Theorem 2 summarizes Theorems 4-6 of De Bortoli et al. (2021), which concern the convergence properties of the SOUL algorithm. An extension to nonconvex $-\log p(y|\theta)$ is given when $\Theta$ is a sufficiently regular manifold in De Bortoli et al. (2021, Appendix B), achieving (a.s.) convergence to some point $\theta^*$ with $-\nabla_\theta \log p(y|\theta^*)$ lying on the normal cone of $\Theta$ at $\theta^*$.

**Theorem 2.** *(De Bortoli et al., 2021, Theorem 4, 5, 6). Assume that Assumptions 1 to 6 hold and $-\log p(y|\theta)$ is convex w.r.t. $\theta$. Assume further that the step-sizes $(\delta_n)_{n \in \mathbb{N}}$ satisfy $\sup_{n \in \mathbb{N}} \delta_n < 1/L_\Phi$, and that $\gamma \leq \min(1, 2m_2)$ where $m_2$ is the tail coercivity constant in Assumption 5. Then for any $n \in \mathbb{N}^*$,*

$$\limsup_{n \to \infty} \mathbb{E}\left[ \max_{\theta \in \Theta} \log p(y|\theta) - \frac{\sum_{k=1}^{n} \delta_k \log p(y|\theta_k)}{\sum_{k=1}^{n} \delta_k} \right] \leq \mathcal{O}(\sqrt{\gamma} + \sqrt{\gamma'}). \tag{17}$$

Theorem 2 bounds the bias of the SAPG method in terms of the step-size of the Markov chains. This covers the Gaussian deconvolution experiments in the next section, but not the Poisson denoising experiments which employ reflected ULA. However, we believe that the SAPG algorithm is also convergent when the Markov kernel is given by reflected ULA, and refer proof of convergence to future work.

We note that almost sure convergence can also be obtained by using decreasing step-sizes or increasing batch-size $m_n$ (De Bortoli et al., 2021, Thm. 1-3). For the sake of simplicity and computational efficiency, we consider only constant step-sizes $\delta_n$ and $\gamma, \gamma'$, and fix $m_n = 1$ for our experiments. For an empirical study into the bias of the SAPG method for low-dimensional (where ground-truth maximum likelihood estimators are known), we refer to (De Bortoli et al., 2021; Vidal et al., 2020). These works find that the bias is not significant if the hyperparameter guidelines are followed, as we do in our experiments.

## 4 Experiments

In the following image reconstruction experiments, we will compare the proposed unsupervised MLE method Algorithm 2 with the $t$-gradient-step supervised training method as detailed in Goujon et al. (2022). We use the same choice of architecture as Goujon et al. (2022) extended from grayscale images to color images. In (14) and (15), $\mathbf{W}$ is modelled using a convolution, and the $\sigma_i$ are parameterized using linear splines. This results in $\psi_i$ being degree-2 splines, namely piecewise quadratic with continuous derivatives. We consider having 32 splines for our parameterization, and $\mathbf{W}$ being convolutions with 3 (color) input channels and 32 output channels. Additional implementation details including the choice of $\mathbf{W}$ can be found in Appendix B.

The supervised gradient-step (GS) training method of Goujon et al. (2022) involves performing gradient steps to minimize the variational functional induced by the convex regularizer, which is then trained to minimize the $\ell_1$ distance between the reconstructed image and the ground truth. We compare the supervised GS training to our proposed unsupervised training (SAPG), which does not require noise-free images.

Equivariant imaging poses another interesting baseline to compare against for unsupervised learning. We use the setup in Scanvic et al. (2023), which recently extends the EI framework using scale transforms, used in cases where the forward operator and noise models are equivariant under the standard rotations and reflections. The learned reconstruction of the EI model takes the form of an end-to-end neural network, using the state-of-the-art SwinIR architecture for image reconstruction, using 11.5M parameters (Liang et al., 2021). We train using the given parameters for up to 200 epochs, but use early stopping at 25 epochs due to divergence issues during training.

For baselines that do not rely on machine learning, we compare with the total variation prior $g(x) = \|\nabla x\|_1$ (Rudin et al., 1992), as well as using a deep image prior (DIP) for reconstruction (Tachella et al., 2021; Ulyanov et al., 2018). For TV regularization, the regularization parameter is chosen via a grid search to maximize PSNR with respect to the ground truth. We use TV to additionally compare with the single-image self-supervised approach of (Vidal et al., 2020), which automatically finds the regularization parameter using MLE. Note that the reported quality for TV will be higher than if (Vidal et al., 2020) is used, due to the comprehensive search. For DIP, we compare using a U-Net as well as a convolutional neural network, optimized using Adam and gradient descent as in Tachella et al. (2021). The reported PSNR is taken to be the highest with respect to the ground truth along the optimization trajectory.

To evaluate our MLE-trained regularizers, we consider the posterior mean and MAP estimators under the posterior chain. The posterior mean is evaluated after $2 \times 10^4$ and $1 \times 10^5$ iterations of the posterior Markov chains with $5 \times 10^3$ iterations to warm-start, labeled 'MMSE' in the figures and tables with superscripts indicating the number of posterior samples. To compute the MAP estimators of our trained regularizers, we minimize the modified negative log-posterior $\varphi_\theta(x) = f_y(x) + \lambda g_\theta(x)$, where $f_y$ is the negative log-likelihood (fidelity) function corresponding to the corrupted data $y$, and $g_\theta$ is the learned convex regularizer, and $\lambda > 0$ is a regularization parameter. The regularization parameter $\lambda$ is chosen via a grid search $\lambda \in \{0.1, 0.2, ..., 1.0\}$ to maximize PSNR after training. To compute the MAP estimate, the negative log-posterior $\varphi_\theta$ is minimized using the Adam optimizer for up to $10^4$ iterations due to being faster than gradient descent, with learning rate $10^{-3}$ and other parameters as default. We report the training and testing time in Appendix C.

To explore the strength of the unsupervised SAPG training method along with the supervised gradient-step method of Goujon et al. (2022), we additionally compared with a weakly-supervised training regime, by warm-starting the regularizer using adversarial training based on Lunz et al. (2018). This uses the formulation (12) with the noisy distribution being corrupted images, and is labelled as WS-SAPG in following tables. The weakly-trained regularizer is first provided ground truths of a fixed 2% of the training images, and the CRR is trained to be maximized on the corresponding corrupted images and minimized on the ground truth. The loss for warm-starting can be expressed as follows, where the last term controls the Lipschitz constant of $g_\theta$.

$$\min_\theta \mathcal{L}_{\text{warmstart}}(\theta) = \frac{1}{M} \sum_{i=1}^{M} \left[ g_\theta(x_i^{\text{true}}) - g_\theta(x_i^{\text{noisy}}) \right] + \max(1, \text{Lip}(g_\theta)). \tag{18}$$

After warm-starting, the adversarially learned CRR is passed as an initialization into the standard SAPG method, where only the corrupted training dataset is provided for use in Algorithm 2.

Table 1: Table of PSNR (dB) of the convex ridge regularizer (CRR) trained for Gaussian deconvolution. The initial PSNR of the corrupted image is 22.38dB and the initial SSIM is 0.612. The MAP regularization parameter for the MLE is $\lambda = 0.6$, and values are averaged over 50 test images. We observe only a small drop in performance of the unsupervised SAPG and weakly-supervised WS-SAPG compared to the supervised gradient-step (GS) method of around 0.5dB, and outperform total variation. Equivariant imaging (EI) performs very well, possibly due to a more powerful neural network parameterization.

| | | CRR-based | | | | | | | | End-to-end/model-based | | | |
| | GS | SAPG | | | WS-SAPG | | | EI | TV | DIP | | | |
| | | MAP | MMSE$^{2e4}$ | MMSE$^{1e5}$ | MAP | MMSE$^{2e4}$ | MMSE$^{1e5}$ | | | UNet-Adam | UNet-GD | CNN-Adam | CNN-GD |
|---|---|---|---|---|---|---|---|---|---|---|---|---|---|
| PSNR | 25.74 | 25.19 | 25.09 | 25.10 | 25.25 | 25.05 | 25.06 | 27.11 | 24.75 | 25.50 | 21.80 | 24.67 | 24.50 |
| SSIM | 0.803 | 0.795 | 0.779 | 0.780 | 0.802 | 0.766 | 0.767 | 0.846 | 0.786 | 0.784 | 0.565 | 0.723 | 0.739 |
| Params | | | | 15,537 | | | | 11,504,163 | 1 | 1,036,032 | | 187,392 | |

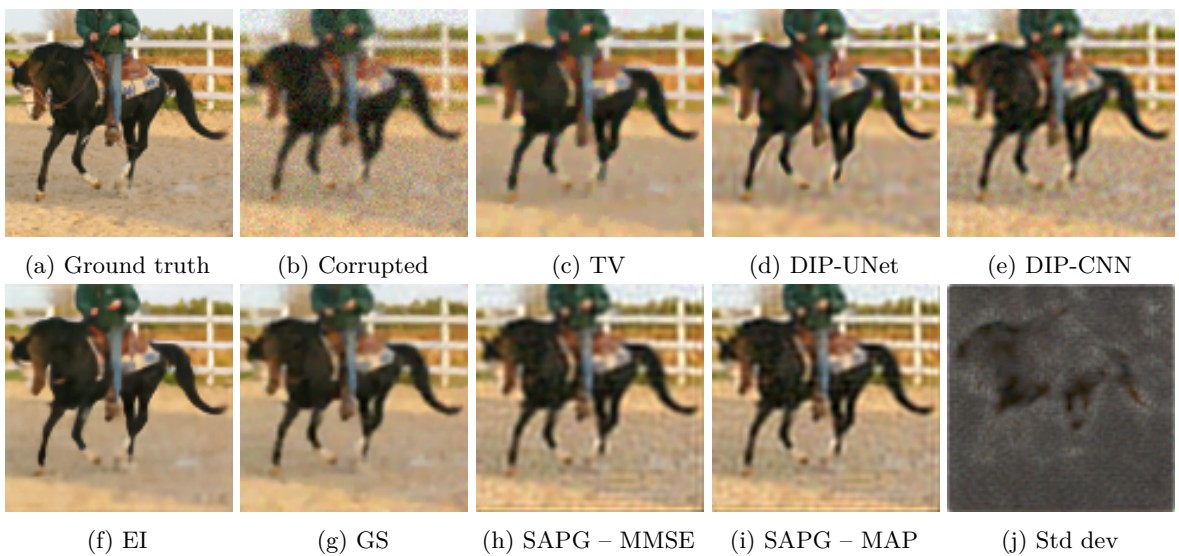

(a) Ground truth     (b) Corrupted     (c) TV     (d) DIP-UNet     (e) DIP-CNN

(f) EI     (g) GS     (h) SAPG – MMSE     (i) SAPG – MAP     (j) Std dev

Figure 1: Visual comparison of various reconstructions of a blurred test image with Gaussian noise. The unsupervised MMSE (h) and MAP (i) reconstructions of the proposed SAPG method contain visual artifacts compared to the gradient-step training method. The standard deviation (j) shows the uncertainty around edges. We observe that EI has the sharpest looking image, which is due to explicit knowledge of the forward operator as a blur kernel, as well as having a much richer end-to-end parameterization.

## 4.1 Gaussian Deconvolution

The first set of experiments is Gaussian deconvolution on the natural image dataset STL-10. The images consist of $96 \times 96$ color images, which are then corrupted with a Gaussian blur with kernel size 5 and blur strength 1, followed by additive 5% Gaussian noise. The negative log-likelihood for noisy blurred image $y$ is thus given by

$$f_y(x) = \frac{1}{2\sigma^2}\|Ax - y\|^2,$$

where $A$ is the blur kernel. For SAPG, the step-sizes $\gamma, \gamma'$ for the likelihood and prior Markov chains $\mathrm{R}_{\gamma,\theta}$, $\bar{\mathrm{R}}_{\gamma',\theta}$ respectively are given by $\gamma = \gamma' = 1e-4$.

Table 1 details the reconstruction quality in terms of PSNR of the proposed SAPG method as well as various supervised and unsupervised baselines. We observe that the proposed unsupervised SAPG method is able to perform closely with the supervised $t$-gradient-step method, with a gap of only 0.5dB. Moreover, we observe that the MAP estimate with the learned reconstruction priors are generally better than the posterior mean estimates. The SAPG-learned prior also is competitive with/better than other priors such as TV or DIP,

Table 2: Table of PSNR (dB) of the convex ridge regularizer trained for Poisson denoising. We observe competitive performance of the learned CRR-based priors compared to end-to-end reconstructions, and only a minor drop in performance between the proposed unsupervised SAPG method and supervised GS method. The initial average PSNR of the corrupted images is 21.16dB and the initial SSIM is 0.641. The MAP regularization parameter for the MLE is $\lambda = 0.6$. Averaged over 50 test images.

| | | CRR-based | | | | | | | End-to-end/model-based | | | | | |
|---|---|---|---|---|---|---|---|---|---|---|---|---|---|---|
| | GS | SAPG | | | WS-SAPG | | | EI | TV | DIP | | | | |
| | | MAP | MMSE$^{2e4}$ | MMSE$^{1e5}$ | MAP | MMSE$^{2e4}$ | MMSE$^{1e5}$ | | | UNet-Adam | UNet-GD | CNN-Adam | CNN-GD |
| PSNR | 28.29 | 27.94 | 26.65 | 27.52 | 28.14 | 27.34 | 27.75 | 28.43 | 24.09 | 26.50 | 21.84 | 26.28 | 26.59 |
| SSIM | 0.858 | 0.866 | 0.813 | 0.840 | 0.878 | 0.836 | 0.846 | 0.861 | 0.722 | 0.818 | 0.575 | 0.783 | 0.799 |

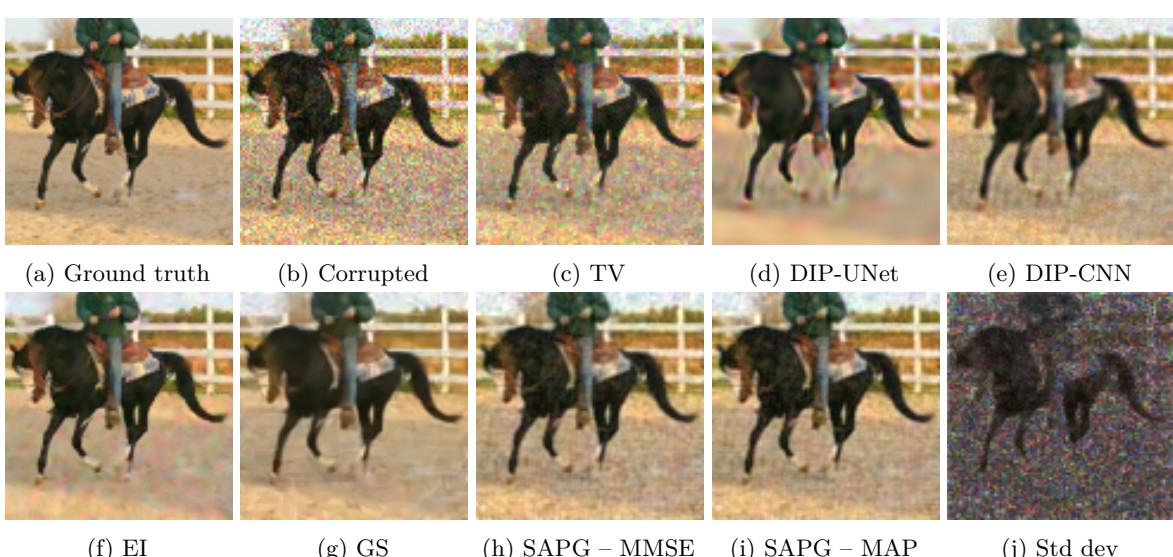

(a) Ground truth    (b) Corrupted    (c) TV    (d) DIP-UNet    (e) DIP-CNN

(f) EI    (g) GS    (h) SAPG – MMSE    (i) SAPG – MAP    (j) Std dev

Figure 2: Visual comparison of reconstructions for Poisson denoising. The proposed unsupervised SAPG method, as shown in subfigures (h) and (i) both show significant denoising, but with the presence of artifacts. EI has color artifacts while the supervised GS method has more textural artifacts. DIP has a strong smoothing effect, but also induces strange visual artifacts around the target as shown in subfigure (d).

the latter of which still has a lack of theoretical properties. Adding weak supervision does not significantly change the performance here.

Figure 1 shows visual comparisons of the various methods applied to a corrupted test image. EI has the best perceptual quality out of the tested images with the powerful end-to-end parameterization. The reconstruction of the proposed method is reasonable, but has artifacts compared to the supervised gradient step method, possibly due to the ill-posedness of the problem. We are also able to plot the standard deviation of the Markov chain samples as seen in Figure 1j, clearly showing areas of interest around the edges.

## 4.2 Poisson Denoising

Poisson denoising naturally arises in low-photon imaging, where the measurement takes integer pixel values. We can model the measurement using a Poisson random variable $y \sim \text{Pois}(\eta x)$ taken pixel-wise over an image space $\mathbb{R}_+^n$, where $\eta$ is a parameter chosen such that the expected mean pixel intensity of $y$ is a fixed mean intensity value (MIV). The forward operator is set to be the identity. In this case, the Poisson negative log-likelihood for positive-integer-valued measurement $y \sim \text{Pois}(\eta x)$ is given as follows, where $\iota_+$ is the

Table 3: Table of PSNR (dB) and drop in PSNR compared to Table 1, of the convex ridge regularizer trained for Gaussian deconvolution when applied to uniform deconvolution. We observe significantly better transfer properties of the learned CRR-based regularizers compared to the end-to-end EI method, with only a small drop in performance between the supervised and unsupervised versions. We also observe better transfer of the unsupervised SAPG training compared to supervised GS training. The initial PSNR of the corrupted image is 20.13dB and the initial SSIM is 0.475. The MAP regularization parameter for the MLE is $\lambda = 0.7$. Averaged over 50 test images.

| | GS | SAPG | | | WS-SAPG | | | EI | TV |
|---|---|---|---|---|---|---|---|---|---|
| | | MAP | MMSE$^{2e4}$ | MMSE$^{1e5}$ | MAP | MMSE$^{2e4}$ | MMSE$^{1e5}$ | | |
| PSNR | 23.90 | 23.73 | 23.62 | 23.55 | 23.79 | 22.81 | 23.05 | 21.37 | 22.20 |
| Drop | 1.84 | 1.46 | 1.47 | 1.55 | 1.46 | 2.24 | 2.01 | 5.74 | 2.55 |
| SSIM | 0.739 | 0.720 | 0.691 | 0.697 | 0.732 | 0.645 | 0.659 | 0.637 | 0.676 |
| Drop | 0.064 | 0.075 | 0.088 | 0.083 | 0.07 | 0.121 | 0.108 | 0.209 | 0.11 |

characteristic function on $\mathbb{R}^n_+$ as in (9), $y!$ denotes factorial, and division of images is to be taken pixel-wise:

$$f_y(x) = \sum_{i=1}^{n} \left[ (\eta x)_i - y_i \log(\eta x_i) + \log(y_i!) \right] + \iota_+(x),$$

$$\nabla_x f_y(x) = \eta \mathbb{1}_{\mathcal{I}} - y/x,$$

The Poisson log-likelihood does not have a globally Lipschitz gradient, degenerating around the boundary of $\mathbb{R}^n_+$ when $x$ is near 0. This makes the Poisson problem more difficult as compared to the Gaussian case due to requiring both approximating the log-likelihood for zero-valued pixels, as well as needing a small step-size. Therefore, we use the following modified negative log-likelihood, for some mollification parameter $b > 0$ (Melidonis et al., 2022),

$$f_y(x) = \sum_{i=1}^{n} \left[ \eta(x_i + b) - y_i \log(\eta(x_i + b)) + \log(y_i!) \right],$$

$$\nabla_x f_y(x) = \eta \mathbb{1}_{\mathcal{I}} - y/(x + b).$$

The introduction of $b > 0$ mollifies the likelihood near zeros and makes $\nabla_x f_y(x)$ Lipschitz on $\mathbb{R}^n_+$. We use the parameter $b = \text{MIV}/100$ as suggested in Melidonis et al. (2022), where the mean intensity value (MIV) is the mean of $y \sim \text{Pois}(\eta x)$. The parameter $\eta$ is chosen such that the mean intensity value is set to be $\lambda_{\text{MIV}} = 25$ for light Poisson denoising. For SAPG, the step-sizes for the likelihood and prior Markov chains are given by $\gamma = 5e{-}6$, $\gamma' = 1e{-}5$ respectively. For EI, the best performance is obtained by modifying the training loss to use the Poisson unbiased risk estimator instead of Stein's unbiased risk estimator, and using rotation transforms instead of scale transforms (Chen et al., 2022a).

Table 2 compares the PSNR of the recovered images using the proposed method against the previously detailed baselines. We again observe a reasonable gap of 0.15dB between our unsupervised method and the supervised method with the same architecture. Figure 2 however still demonstrates the presence of some visual artifacts of our unsupervised method when compared to its supervised $t$-gradient-step counterpart. Moreover, the CRR-based methods exhibit patchy artifacts that are similar to the ground-truth image, while EI exhibits varying color artifacts. For this problem, we achieve much closer performance to equivariant imaging compared to Gaussian deconvolution. This highlights the difficulty of Poisson denoising for end-to-end models, as observed previously due to the high Lipschitz constants and large data range, and an empirical benefit of variational regularization when dealing with high-variance noise.

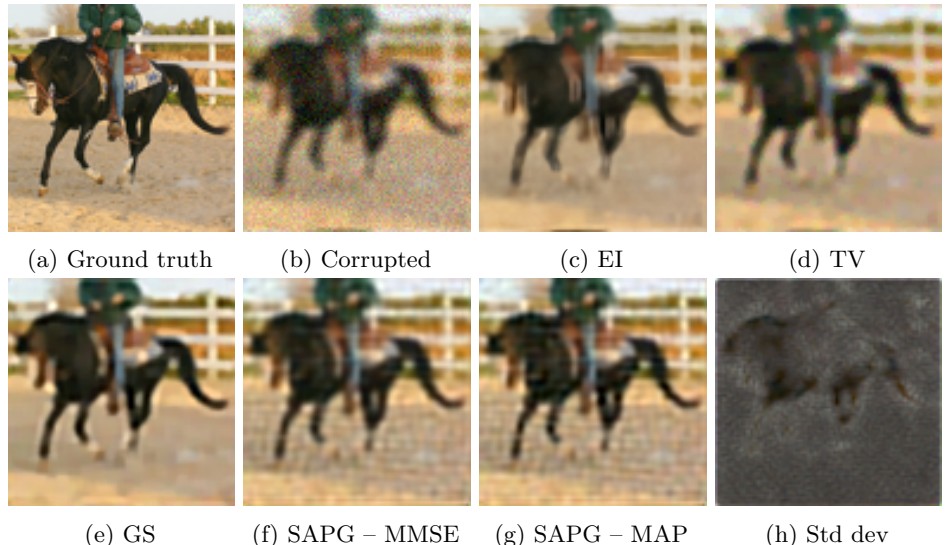

(a) Ground truth (b) Corrupted (c) EI (d) TV

(e) GS (f) SAPG – MMSE (g) SAPG – MAP (h) Std dev

Figure 3: Visual comparison of reconstructions for uniform deconvolution when using models trained for Gaussian deconvolution. We observe more residual blur for EI when compared to the model-based priors in the bottom row.

### 4.3 Transfer to Different Forward Operator

In addition to the Gaussian deconvolution experiments, we consider robustness of the learned models with respect to changes in forward operator. We include the change in forward operator to provide some empirical justification to the choice of learned regularizer as opposed to end-to-end methods. We evaluate the trained models on a $5 \times 5$ uniform blur kernel with the same additive 5% Gaussian noise. In particular, we consider supervised and unsupervised CRR, as well as the end-to-end EI method. The models trained for Gaussian deconvolution in Section 4.1 are applied directly to uniform deconvolution, allowing for a change of forward operator in the fidelity for the variational model-based CRR methods.

Table 3 shows the reconstruction results for transferring the learned models between different tasks. Compared to the learned regularizers and TV, we observe a significantly larger drop in performance from the EI model, which is trained on a specific forward operator, which introduces more blur artifacts. This may be due to the sensitivity of end-to-end networks to changes in distribution. On the other hand, the model-based gradient step reconstruction still produces reasonably good results, due to the underlying variational form. The unsupervised versions have similar performance, albeit with added visual artifacts. This may be due to image prior components that are difficult to learn from only noisy patches.

We note that within learning convex ridge regularizers, our SAPG method drops less PSNR than the supervised gradient-step method when changing forward operator, suggesting better generalization. While both methods utilize the forward operator during training, the gradient-step training unrolls through 10 optimization iterations, whereas the unsupervised method uses it as a consistency term. One possible explanation is that the Langevin diffusion in the unsupervised method expands the available data to also cover some small change in forward operator, while the supervised method sees only data that is provided from the original forward operator.

We additionally note that better generalization under change of forward operator can be achieved using unrolling schemes, which allow for the forward operator to be explicitly specified during the testing phase (Monga et al., 2021; Zhao et al., 2023). Such unrolling methods are typically trained in the supervised regime (Li et al., 2020b). While we found that a direct application of the unsupervised end-to-end EI loss in Scanvic et al. (2023) to an unrolled architecture led to unsatisfactory results in our deconvolution experiments, a more nuanced approach to constructing more transferable architectures could be an interesting direction for future work.

## 5 Conclusion

We proposed an unsupervised method of training a convex neural network-based image prior based on the stochastic approximate proximal gradient algorithm, extending previous work in both model expressiveness and dataset size. Prior works consider a family of total variation priors, here we consider a parameterization with significantly more parameters. Experiments demonstrate that the proposed method produces priors that are near competitive when compared to the alternative supervised training method for various image corruption operators, maintaining significantly better generalization properties when compared to end-to-end methods. Moreover, we present convergence theory for the proposed Markov chains with the choice of convex ridge regularizer architecture.

The proposed method has two main limitations, namely runtime for training and inference, as well as the convexity assumption on the regularizer. The runtime is mainly due to slow convergence of the MCMC and additional stochasticity when training the CRR. A detailed comparison of the train and test runtimes of the compared methods is given in Appendix C, where our method takes an order of magnitude longer. The convexity assumption of the regularizer allows for convergence of SAPG, but may be restrictive in terms of expressiveness.

Interesting future works could include a convergence analysis in the discrete case or with accelerated versions of the training algorithm such as Nesterov acceleration or Adam instead of SGD, or alternative Markov kernels like SK-ROCK instead of ULA (Pereyra et al., 2020). Other possible directions are to add an informative prior on $\theta$, which may change the sampling dynamics, or devising fully unsupervised methods for warm-starting training. We also believe that the strong convexity assumption of the convex ridge regularizer can be dropped, due to the simple structure and coercive tails. A promising alternative is weak convexity, which has been shown to significantly outperform similar convex models (Shumaylov et al., 2023). Future work could include theoretical analysis of the SAPG method in this setting using techniques from Durmus & Moulines (2017); De Bortoli et al. (2021). Another interesting work would be to extend this work to learning diffusion models from single copies of noisy measurements.

### Acknowledgments

HYT was supported by GSK.ai and the Masason Foundation. MP was funded by the UK Research and Innovation (UKRI) Engineering and Physical Sciences Research Council (EPSRC) through grants EP/T007346/1, EP/V006134/1 and EP/W007673/1. CBS acknowledges support from the Philip Leverhulme Prize, the Royal Society Wolfson Fellowship, the EPSRC advanced career fellowship EP/V029428/1, EPSRC Grants EP/S026045/1 and EP/T003553/1, EP/N014588/1, EP/T017961/1, the Wellcome Innovator Awards 215733/Z/19/Z and 221633/Z/20/Z, the European Union Horizon 2020 research and innovation programme under the Marie Skłodowska-Curie Grant agreement No. 777826 NoMADS, the Cantab Capital Institute for the Mathematics of Information and the Alan Turing Institute.

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

# A Theorems and Proofs

## A.1 Required Assumptions

We begin with stating some basic assumptions. These correspond to A1-3 in De Bortoli et al. (2021).

**Assumption 1.** *(Compact domain.)* $\Theta$ *is a convex compact set and* $\Theta \subset \overline{\mathrm{B}}(0, R_\Theta)$ *with* $R_\Theta > 0$.

**Assumption 2.** *(Lipschitz gradient.)* *There exists an open set* $U \subset \mathbb{R}^{d_\Theta}$ *and* $L_\Phi \geq 0$ *such that* $\Theta \subset U$, $\log p(y|\cdot) \in \mathcal{C}^1(\mathrm{U}, \mathbb{R})$, *and for any* $\theta_1, \theta_2 \in \Theta$,

$$\|\nabla_\theta \log p(y|\theta_1) - \nabla_\theta \log p(y|\theta_2)\| \leq L_\Phi \|\theta_1 - \theta_2\|.$$

**Assumption 3.** *For any* $\theta \in \Theta$, *there exist* $H_\theta, \bar{H}_\theta : \mathbb{R}^d \to \mathbb{R}^{d_\Theta}$ *and two probability distributions* $\pi_\theta, \bar{\pi}_\theta$ *on the Borel space* $\left(\mathbb{R}^d, \mathcal{B}\left(\mathbb{R}^d\right)\right)$ *satisfying for any* $\theta \in \Theta$

$$\nabla_\theta \log p(y|\theta) = \int_{\mathbb{R}^d} H_\theta(x) \mathrm{d}\pi_\theta(x) + \int_{\mathbb{R}^d} \bar{H}_\theta(x) \mathrm{d}\bar{\pi}_\theta(x).$$

*In addition,* $(\theta, x) \mapsto H_\theta(x)$ *and* $(\theta, x) \mapsto \bar{H}_\theta(x)$ *are measurable.*

Assumptions 1 and 2 are standard compactness and Lipschitz assumptions in the literature, and we note that Assumption 3 holds if Fisher's identity (6) holds. We additionally require some assumptions to guarantee ergodicity of the two Markov chains in Equation (7). The following assumptions correspond to L1-3 in De Bortoli et al. (2021).

**Assumption 4.** *(Uniformly Lipschitz gradients.)* *For any* $\theta \in \Theta$, $\bar{\pi}_\theta$ *admits a probability density function with respect to to the Lebesgue measure proportional to* $x \mapsto \exp(-g_\theta(x))$ *and* $\pi_\theta$ *admits a probability density function with respect to to the Lebesgue measure proportional to* $x \mapsto \exp(-E_\theta(x))$. *In addition,* $(\theta, x) \mapsto g_\theta(x)$ *and* $(\theta, x) \mapsto E_\theta(x)$ *are continuous,* $x \mapsto g_\theta(x)$ *and* $x \mapsto E_\theta(x)$ *are differentiable for all* $\theta \in \Theta$ *and there exists* $L_g \geq 0$, $L_E \geq 0$ *such that for any* $x_1, x_2 \in \mathbb{R}^d$,

$$\sup_{\theta \in \Theta} \|\nabla_x g_\theta(x_1) - \nabla_x g_\theta(x_2)\| \leq L_g \|x_1 - x_2\|$$
$$\sup_{\theta \in \Theta} \|\nabla_x E_\theta(x_1) - \nabla_x E_\theta(x_2)\| \leq L_E \|x_1 - x_2\|$$

*and* $\{\|\nabla_x g_\theta(0)\| : \theta \in \Theta\}$, $\{\|\nabla_x E_\theta(0)\| : \theta \in \Theta\}$ *are bounded.*

**Assumption 5.** *(Tail condition.)* *There exists* $m_1 > 0$ *and* $m_2, \mathrm{c}, R_1 \geq 0$ *such that for any* $\theta \in \Theta$ *and* $x \in \mathbb{R}^d$,
$$\langle \nabla_x E_\theta(x), x \rangle \geq \mathrm{m}_1 \|x\| \mathbb{1}_{\mathbb{R}^d \setminus \mathrm{B}(0, R_1)}(x) + \mathrm{m}_2 \|\nabla_x E_\theta(x)\|^2 - \mathrm{c}.$$

**Assumption 6.** *(Well-conditioned w.r.t.* $\theta$*.)* *There exist constants* $\mathrm{M}_{1,\Theta} \geq 0$ *and* $\mathrm{M}_{2,\Theta} \geq 0$ *such that for any* $\theta_1, \theta_2 \in \Theta, x \in \mathbb{R}^d$,

$$\|\nabla_x E_{\theta_1}(x) - \nabla_x E_{\theta_2}(x)\| \leq \mathrm{M}_{1,\Theta} \|\theta_1 - \theta_2\| \exp\left(\mathrm{m}_1 \sqrt{1 + \|x\|^2}/8\right),$$
$$\|\nabla_x g_{\theta_1}(x) - \nabla_x g_{\theta_2}(x)\| \leq \mathrm{M}_{2,\Theta} \|\theta_1 - \theta_2\| \exp\left(\mathrm{m}_1 \sqrt{1 + \|x\|^2}/8\right).$$

Assumption 4 is a necessary condition for the SDEs corresponding to the ULA Algorithms 1 and 2 to have unique solutions. Assumptions 5 and 6 provide conditions for geometric ergodicity of ULA, which asserts geometric convergence to the invariant distribution in total variation (Meyn & Tweedie, 1992). Other sufficient conditions for geometric ergodicity of ULA can be found in Durmus & Moulines (2017). We note that the exponential term in Assumption 6 is not necessary, and we show a tighter bound growing linearly in $\|x\|$ similar to that used in (De Bortoli et al., 2020).

We provide here some intuition into how each condition is satisfied by the convex ridge regularizer. Detailed proofs are left to the appendix.

**Proposition 1.** *Suppose $g_\theta$ takes the form of a convex ridge regularizer (14), where the convex profile functions $\psi_i$ are $\mathcal{C}^1$, parameterized using piecewise quadratic splines, and the parameters $\theta$ takes values in some compact $\Theta$. Suppose further that $f_y(x)$ is convex and also has Lipschitz gradient, and $\ell(y|x) = \exp(-f_y(x))$ has finite second moment. Then Assumptions 1 to 6 all hold.*

*Proof sketch. Assumption 1*: The compact $\Theta$ can be chosen a-priori to be sufficiently large to allow for optimization.

*Assumption 2*: Recalling the form of the CRR, $g_\theta(x) : \mathbf{x} \mapsto \sum_{i=1}^{C} \psi_i(\mathbf{w}_i^\top \mathbf{x})$, this is Lipschitz on compact sets by chain rule. The dominated convergence theorem together with Fisher's identity transfers this to the log marginal $\nabla_\theta p(y|\theta)$. See Appendix B.1, which writes the derivative of the CRR in closed form.

*Assumption 3*: Follows from Fisher's identity (6). The required conditions for Fisher's identity follow (Vidal et al., 2020, Prop. A.1.), verified by the CRR:

1. For any $\theta \in \Theta$ and $\tilde{x} \in \mathbb{R}^d$, $(x, y) \mapsto p(x, y|\theta), y \mapsto p(y|\tilde{x})$ are positive p.d.f.s.

   - Follows since $f_y, g_\theta$ are finite and convex, and that $g_\theta$ is constructed to grow quadratically for large $x$.

2. For any $x \in \mathbb{R}^d, \theta \in \text{int}(\Theta)$, $\theta \mapsto p(y, x|\theta)$ is differentiable.

   - Follows since CRRs are continuously differentiable w.r.t. $\theta$ for fixed $x$.

3. For any $y \in \mathbb{R}^{d_y}$ and $\theta \in \text{int}(\Theta)$, there exists $\epsilon > 0$ and measurable $\tilde{g}$ such that for any $\tilde{\theta} \in \bar{B}_\epsilon(\theta)$ and $x \in \mathbb{R}^d$, $\|\nabla_\theta \log p(y, x|\tilde{\theta})\| \leq \tilde{g}(x)$ with $\int \tilde{g}(x) p(y|x) \mathrm{d}x < +\infty$.

   - Since $\Theta$ is compact, $\|\nabla_\theta \log p(y, x|\tilde{\theta})\| = \|\nabla_\theta g_\theta(x)\|$ can be bounded by $c\|x\|^2$ as in Appendix B.1, which is integrable under $p(y|x)$ by assumption of having finite second moment.

We have $\pi_\theta = p(x|\theta)$, $\bar{\pi}_\theta = p(x|y, \theta)$, and $H_\theta = -\bar{H}_\theta = \nabla_\theta g_\theta$. This is also given in De Bortoli et al. (2020, Remark 2).

*Assumption 4*: Since $\Theta$ is compact and $g_\theta$ are piecewise quadratic, the ridge regularizers have uniformly Lipschitz gradient (Goujon et al., 2022, Prop. IV.1.). The condition for $E_\theta$ holds as well since $f_y(x)$ has Lipschitz gradient. See Lemma 1.

*Assumption 5*: For simplicity, we can enforce the convex ridge regularizer to strongly convex by a choice of spline parameters. In this case, $E_\theta = f_y + g_\theta$ is also strongly convex, and thus for sufficiently large $x$, is bounded below by $c_1\|x\|^2$. Apply Young's inequality and taking sufficiently large $c, R_1$ for the desired tail condition. In general, we need only strong monotonicity of the gradient for sufficiently large $x$ w.r.t. the origin. See Lemma 2.

*Assumption 6*: By continuity of the convex ridge regularizer, and that both $g_\theta$ and $E_\theta$ grow at most quadratically in $\|x\|$. See Lemma 3. $\qquad\square$

This rest of this section is divided into three subsections for clarity, addressing Assumptions 4-6. Recall the definition of the CRR and the form of its gradient:

$$g_\theta : \mathbf{x} \mapsto \sum_{i=1}^{C} \psi_i(\mathbf{w}_i^\top \mathbf{x}), \quad \nabla g_\theta(\mathbf{x}) = \mathbf{W}^\top \sigma(\mathbf{W}\mathbf{x}), \tag{19}$$

where $\psi_i : \mathbb{R} \to \mathbb{R}$ are convex profile functions (ridges) applied element-wise, and $\mathbf{w}_i \in \mathbb{R}^d$ are learnable weights, $\mathbf{W} = [\mathbf{w}_1 \dots \mathbf{w}_C]^\top \in \mathbb{R}^{C \times d}$ is linear, and $\sigma : \mathbb{R}^C \to \mathbb{R}^C$ is a component-wise activation function with components $\sigma_i = \psi_i' : \mathbb{R} \to \mathbb{R}$.

## A.2  Assumption 4 for CRRs

**Assumption 4.** *(Uniformly Lipschitz gradients.) For any $\theta \in \Theta$, $\bar{\pi}_\theta$ admits a probability density function with respect to to the Lebesgue measure proportional to $x \mapsto \exp(-g_\theta(x))$ and $\pi_\theta$ admits a probability density function with respect to the Lebesgue measure proportional to $x \mapsto \exp(-E_\theta(x))$. In addition, $(\theta, x) \mapsto g_\theta(x)$ and $(\theta, x) \mapsto E_\theta(x)$ are continuous, $x \mapsto g_\theta(x)$ and $x \mapsto E_\theta(x)$ are differentiable for all $\theta \in \Theta$ and there exists $L_g \geq 0$, $L_E \geq 0$ such that for any $x_1, x_2 \in \mathbb{R}^d$,*

$$\sup_{\theta \in \Theta} \|\nabla_x g_\theta(x_1) - \nabla_x g_\theta(x_2)\| \leq L_g \|x_1 - x_2\|$$

$$\sup_{\theta \in \Theta} \|\nabla_x E_\theta(x_1) - \nabla_x E_\theta(x_2)\| \leq L_E \|x_1 - x_2\|$$

*and $\{\|\nabla_x g_\theta(0)\| : \theta \in \Theta\}$, $\{\|\nabla_x E_\theta(0)\| : \theta \in \Theta\}$ are bounded.*

**Lemma 1.** *Convex ridge regularizers Equation* (14) *satisfy Assumption 4.*

*Proof.* From Equation (15) and Appendix B, since $\Theta$ is compact and bounded, the slope of $\sigma$ is uniformly bounded and hence $\sigma$ is uniformly Lipschitz, say with constant $L_\sigma > 0$. Moreover, $\|W\|$ is uniformly bounded (in operator norm). Given $x_1, x_2 \in \mathbb{R}^d$,

$$\|\nabla_x g_\theta(x_1) - \nabla_x g_\theta(x_2)\| \leq \|\mathbf{W}^\top \sigma(\mathbf{W}x_1) - \mathbf{W}^\top \sigma(\mathbf{W}x_2)\| \leq \|\mathbf{W}\| L_\sigma \|\mathbf{W}\| \|x_1 - x_2\|$$

Let $L_g = L_\sigma \sup_{\theta \in \Theta} \left(\|\mathbf{W}\|^2\right)$. Then $L_g < +\infty$ and we complete the inequalities with $L_E = L_g + L_f$, where both $\nabla_x g_\theta$ and thus $E_\theta$ are uniformly Lipschitz with respect to $x$, over $\theta \in \Theta$. The rest follows from compactness of $\Theta$ and the continuous parameterization of the CRRs. $\square$

## A.3  Assumption 5 for CRRs

**Assumption 5.** *(Tail condition.) There exists $m_1 > 0$ and $m_2, c, R_1 \geq 0$ such that for any $\theta \in \Theta$ and $x \in \mathbb{R}^d$,*

$$\langle \nabla_x E_\theta(x), x \rangle \geq m_1 \|x\| \mathbb{1}_{\mathbb{R}^d \setminus B(0, R_1)}(x) + m_2 \|\nabla_x E_\theta(x)\|^2 - c.$$

**Lemma 2.** *Convex ridge regularizers Equation* (14) *satisfy Assumption 5.*

*Proof.* Convex ridge regularizers are strongly convex, which can be enforced by a choice of spline parameters, see Goujon et al. (2022) or Appendix B. Therefore, given $\theta \in \Theta$, let $E_\theta$ have strong convexity constant $m_\theta > m > 0$ bounded below by $m$. Given $x, x' \in \mathbb{R}^d$, $E_\theta$ satisfies

$$\langle x - x', \nabla E_\theta(x) - \nabla E_\theta(x') \rangle \geq m_\theta \|x - x'\|^2. \tag{20}$$

From Lemma 1, $\nabla E_\theta(\cdot)$ is uniformly $L_E$-Lipschitz over $\theta \in \Theta$. For all $x, x' \in \mathbb{R}^d$, we have the following inequalities (Nesterov, 2013, Thm. 2.1.5):

$$\langle x - x', \nabla E_\theta(x) - \nabla E_\theta(x') \rangle \leq L_E \|x - x'\|^2, \tag{21}$$

$$\|\nabla E_\theta(x) - \nabla E_\theta(x')\|^2 \leq L_E \langle x - x', \nabla E_\theta(x) - \nabla E_\theta(x') \rangle. \tag{22}$$

We compute as follows:

$$\|\nabla E_\theta(x)\|^2 \leq 2 \|\nabla E_\theta(x) - \nabla E_\theta(x')\|^2 + 2 \|\nabla E_\theta(x')\|^2 \qquad \text{parallelogram law}$$

$$\leq 2L_E \langle x - x', \nabla E_\theta(x) - \nabla E_\theta(x') \rangle + 2 \|\nabla E_\theta(x')\|^2 \qquad \text{by (22)}.$$

Rearranging, we have

$$\langle x - x', \nabla E_\theta(x) - \nabla E_\theta(x')\rangle \geq \frac{1}{2L_E} \|\nabla E_\theta(x)\|^2 - \frac{1}{L_E} \|\nabla E_\theta(x')\|^2.$$

We combine this with (20) by halving both to obtain

$$\langle x - x', \nabla E_\theta(x) - \nabla E_\theta(x')\rangle \geq \frac{m_\theta}{2} \|x - x'\|^2 + \frac{1}{4L_E} \|\nabla E_\theta(x)\|^2 - \frac{1}{2L_E} \|\nabla E_\theta(x')\|^2. \tag{23}$$

Now let $x' = 0$. For any $x$, we have the following by rearranging.

$$\langle x, \nabla E_\theta(x) - \nabla E_\theta(0)\rangle \geq \frac{m_\theta}{2} \|x\|^2 + \frac{1}{4L_E} \|\nabla E_\theta(x)\|^2 - \frac{1}{2L_E} \|\nabla E_\theta(0)\|^2$$

$$\Rightarrow \langle x, \nabla E_\theta(x)\rangle \geq \frac{m_\theta}{2} \|x\|^2 + \frac{1}{4L_E} \|\nabla E_\theta(x)\|^2 - \frac{1}{2L_E} \|\nabla E_\theta(0)\|^2 + \langle x, \nabla E_\theta(0)\rangle$$

$$\geq \frac{m_\theta}{2} \|x\|^2 - \|x\| \|\nabla E_\theta(0)\| + \frac{1}{4L_E} \|\nabla E_\theta(x)\|^2 - \frac{1}{2L_E} \|\nabla E_\theta(0)\|^2, \tag{24}$$

where the final inequality follows from Cauchy-Schwarz.

For $\|x\| \leq 4\|\nabla E_\theta(0)\|/m_\theta$, we have from (24) that

$$\langle x, \nabla E_\theta(x)\rangle \geq \frac{1}{4L_E} \|\nabla E_\theta(x)\|^2 - \left(\frac{4}{m_\theta} + \frac{1}{2L_E}\right) \|\nabla E_\theta(0)\|^2.$$

For $\|x\| \geq 4\|\nabla E_\theta(0)\|/m_\theta$, we have

$$\langle x, \nabla E_\theta(x)\rangle \geq \frac{m_\theta}{2} \frac{4}{m_\theta} \|\nabla E_\theta(0)\| \|x\| - \|x\| \|\nabla E_\theta(0)\| + \frac{1}{4L_E} \|\nabla E_\theta(x)\|^2 - \frac{1}{2L_E} \|\nabla E_\theta(0)\|^2$$

$$\geq \|\nabla E_\theta(0)\| \|x\| + \frac{1}{4L_E} \|\nabla E_\theta(x)\|^2 - \frac{1}{2L_E} \|\nabla E_\theta(0)\|^2.$$

Thus we have

$$\langle x, \nabla E_\theta(x)\rangle \geq \|\nabla E_\theta(0)\| \|x\| \mathbb{1}_{B(0, 4\|\nabla E_\theta(0)\|/m_\theta)^c}(x) + \frac{1}{4L_E} \|\nabla E_\theta(x)\|^2 - \left(\frac{4}{m_\theta} + \frac{1}{2L_E}\right) \|\nabla E_\theta(0)\|^2. \tag{25}$$

We conclude by noting that $m_\theta > m > 0$ is lower-bounded and that the inequalities continue to hold if the ball radius is taken to be $R_1 = \sup_\Theta 4\|\nabla E_\theta(0)\|/m$, which is finite as $\Theta$ is compact. $\qquad\square$

### A.4 Assumption 6 for CRRs

**Assumption 6.** *(Well-conditioned w.r.t. $\theta$.) There exist constants $M_{1,\Theta} \geq 0$ and $M_{2,\Theta} \geq 0$ such that for any $\theta_1, \theta_2 \in \Theta, x \in \mathbb{R}^d$,*

$$\|\nabla_x E_{\theta_1}(x) - \nabla_x E_{\theta_2}(x)\| \leq M_{1,\Theta} \|\theta_1 - \theta_2\| \exp\left(m_1 \sqrt{1 + \|x\|^2}/8\right),$$

$$\|\nabla_x g_{\theta_1}(x) - \nabla_x g_{\theta_2}(x)\| \leq M_{2,\Theta} \|\theta_1 - \theta_2\| \exp\left(m_1 \sqrt{1 + \|x\|^2}/8\right).$$

**Lemma 3.** *Convex ridge regularizers Equation (14) satisfy Assumption 6.*

*Proof.* From Equation (15), since $\Theta$ is compact and bounded, the slope of $\sigma$ is bounded and $\sigma$ is uniformly Lipschitz, say with Lipschitz constant bounded by $L_\sigma$.

For any $\theta_1, \theta_2 \in \Theta$ and $x \in \mathbb{R}^d$, assume that $\mathbf{W}_1$ is a part of $\theta_1$ and $\mathbf{W}_2$ is a part of $\theta_2$, then we have

$$
\begin{aligned}
\|\nabla_x g_{\theta_1}(x) - \nabla_x g_{\theta_2}(x)\| &\leq \left\| \mathbf{W}_1^\top \sigma_1(\mathbf{W}_1 x) - \mathbf{W}_2^\top \sigma_2(\mathbf{W}_2 x) \right\| \\
&\leq \left\| \mathbf{W}_1^\top \sigma_1(\mathbf{W}_1 x) - \mathbf{W}_1^\top \sigma_1(\mathbf{W}_2 x) \right\| + \left\| \mathbf{W}_1^\top \sigma_1(\mathbf{W}_2 x) - \mathbf{W}_1^\top \sigma_2(\mathbf{W}_2 x) \right\| + \left\| \mathbf{W}_1^\top \sigma_2(\mathbf{W}_2 x) - \mathbf{W}_2^\top \sigma_2(\mathbf{W}_2 x) \right\| \\
&\leq L_\sigma \|\mathbf{W}_1\| \|\mathbf{W}_1 - \mathbf{W}_2\| \|x\| + 2 L_\sigma \|\mathbf{W}_1\| \|\mathbf{W}_2\| \|x\| + L_\sigma \|\mathbf{W}_1 - \mathbf{W}_2\| \|\mathbf{W}_2\| \|x\| \\
&\leq \mathrm{M}_{2,\Theta} \|\theta_1 - \theta_2\| \|x\|,
\end{aligned}
$$

where

$$
\mathrm{M}_{2,\Theta} = L_\sigma \sup_{\theta_1, \theta_2 \in \Theta} \left( \|\mathbf{W}_1\| \|\mathbf{W}_1 - \mathbf{W}_2\| + 2 \|\mathbf{W}_1\| \|\mathbf{W}_2\| + \|\mathbf{W}_1 - \mathbf{W}_2\| \|\mathbf{W}_2\| \right) < +\infty. \tag{26}
$$

Conclude the second inequality by using Assumption 5 and noting $1 + \|x\| < C \exp\left( \mathrm{m}_1 \sqrt{1 + \|x\|^2}/8 \right)$ for some sufficiently large $C$. The first inequality comes from $f_y$ having Lipschitz gradient. $\qquad \square$

## B  Convex Ridge Regularizer Implementation

In this section, we detail the convex ridge regularizer architecture, and provide a closed-form derivative with respect to its parameters in Appendix B.1. Recall from Section 4 that the convex ridge regularizer takes the form

$$
R : \mathbf{x} \mapsto \sum_i \psi_i(\mathbf{w}_i^\top \mathbf{x}), \tag{27}
$$

where $\mathbf{W} = [\mathbf{w}_1 \dots \mathbf{w}_C]^\top \in \mathbb{R}^{C \times d}$ with learnable weights $\mathbf{w}_i \in \mathbb{R}^d$, and $\psi_i : \mathbb{R} \to \mathbb{R}$ are convex profile functions (ridges). In practice, we use matrices for $\mathbf{w}_i$, so that $\mathbf{w}_i^\top x$ represents a vector, and apply $\psi_i$ component-wise. This can be justified by using duplicate $\psi_i$ and weight-tying.

**Choice of W.** As detailed in Section VI.A in Goujon et al. (2022), $\mathbf{W}$ is taken as the composition of two zero-padded convolutions with window size $7 \times 7$, with 8 and 32 output channels respectively. While the experiments in Goujon et al. (2022) are originally in black-and-white, we extend to color images by using 3 input channels instead of 1 for the first convolution.

**Choice of ridges $\psi_i$.** As in Goujon et al. (2022), we parameterize each component's derivative $\sigma_i = \psi_i'$ : $\mathbb{R} \to \mathbb{R}$ as a linear spline, and subsequently define the convex profile function as an integral $\psi_i(t) = \int_0^t \sigma_i(s)\, \mathrm{d}s$. To parameterize the linear splines $\sigma_i$, we define the values at equispaced knots, clamping during training to maintain strictly increasing slopes, ensuring strong convexity of the ridge $\psi_i$. We use 21 equally distant knots (centered at 0), with distance $\Delta = 0.01$ between them. Values defined between knots are given by linear interpolation, and values past the endpoints are given by linear extension. In particular, we have $\sigma_i(t) \sim c_{+,i} t$ as $t \to +\infty$ and $\sigma_i(t) \sim c_{-,i} t$ as $t \to -\infty$ for some $c_{\pm,i} > 0$.

By clamping the allowable knots, we get that for a compact parameter set $\Theta$, $\sigma_i$ are uniformly Lipschitz, used in Lemma 1.

### B.1  Gradient of the CRR

We justify Assumption 2 and the third claim for Assumption 3 in Proposition 1 by computing the derivative of $g_\theta$ with respect to its parameters $\theta$.

Fix a CRR architecture, where the linear spline $\sigma_i = \psi_i'$ has (fixed) knots at $t_{-K}, \dots, t_K$, taking (learnable) values $c_{-K}, \dots, c_K$. We assume that the knots are equispaced $t_k = k\Delta$ for $k = -K, \dots, K$, that $c_0 = 0$, relaxing these assumptions are simple and do not significantly change the proof. For strong convexity, we require that the sequence $c_{-K}, \dots, c_K$ is strictly increasing. We can thus reparameterize in terms of spline differences $d_{-K}, \dots, d_K$ that are lower bounded by some constant $m > 0$ to enforce strong convexity:

$$
\sigma_i(t_k) = c_k = \begin{cases} \sum_{j=1}^k d_j & k > 0 \\ -\sum_{j=k}^{-1} d_j & k < 0 \,, \quad d_k \geq m > 0 \text{ for } k = -K, \dots, -1, 1, \dots, K. \\ 0 & k = 0 \end{cases} \tag{28}
$$

As we use linear interpolation between the knots (and extrapolation outside), this can be rewritten as a sum of ReLUs:

$$
\sigma_i(t) = \begin{cases} \sum_{j=1}^{k}(d_j - d_{j-1})\max(0, t - t_{j-1})/\Delta & t \in [t_k, t_{k+1}),\ k \geq 0 \\ -\sum_{j=k}^{-1}(d_j - d_{j+1})\max(0, t_{j+1} - t)/\Delta & t \in (t_{k-1}, t_k],\ k \leq 0 \\ 0 & k = 0 \end{cases}
$$

$$
= \begin{cases} \sum_{j=1}^{K}(d_j - d_{j-1})\max(0, t - t_{j-1})/\Delta & t \geq 0 \\ -\sum_{j=-K}^{-1}(d_j - d_{j+1})\max(0, t_{j+1} - t)/\Delta & t \leq 0 \end{cases}.
$$

We can then compute $\psi_i(t) = \int_0^t \sigma_i(t)$. We note that since $\sigma_i$ is a sum of ReLUs (and vice versa in the negative direction), the integral $\psi_i(t)$ is a sum of rectified quadratic units $\mathrm{ReLU}^2(t) = t^2 \mathbf{1}_{t \geq 0}$:

$$
\psi_i(t) = \begin{cases} \sum_{j=1}^{K}(d_j - d_{j-1})\mathrm{ReLU}^2(t - t_{j-1})/\Delta & t \geq 0 \\ -\sum_{j=-K}^{-1}(d_j - d_{j+1})\mathrm{ReLU}^2(t_{j+1} - t)/\Delta & t \leq 0 \end{cases}. \tag{29}
$$

Given $y \in \mathbb{R}^{d_y}$, fix any $\tilde{\theta} \in \mathrm{int}(\Theta)$ and $x \in \mathbb{R}^d$. The parameters $\tilde{\theta}$ consist of the weights $\mathbf{w}_i$ and spline values $c_{-K,i}, ..., c_{K,i}$ corresponding to spline $\sigma_i = \psi_i'$. In the following, we drop the index $i$ for notational convenience. We can compute derivatives with respect to spline parameters as follows. Suppose $\mathbf{w}^\top x \in [t_l, t_{l+1})$, where we define $t_{K+1} = +\infty$. We can differentiate with respect to the spline parameters using (29)

$$
(\partial/\partial_{d_j})\psi(\mathbf{w}^\top x) = \frac{1}{\Delta}\left[\mathrm{ReLU}^2(\mathbf{w}^\top x - t_{j-1}) - \mathrm{ReLU}^2(\mathbf{w}^\top x - t_j)\right]. \tag{30}
$$

Similar computations hold if instead $\mathbf{w}^\top x < 0$, and define $t_{-K-1} = -\infty$. Note that these bounds also hold uniformly in a small ball around $\tilde{\theta}$, as $\mathbf{w}^\top x$ is continuous in $\mathbf{w}$. Moreover, this grows linearly in $\mathbf{w}^\top x$. We can also differentiate with respect to the weight vector $\mathbf{w}$ and get a sum of ReLU units, which again grows linearly in $x$: assuming $\mathbf{w}^\top x$ is positive,

$$
\nabla_{\mathbf{w}}\psi(\mathbf{w}^\top x) = \frac{1}{\Delta}\sum_{j=1}^{K}(d_j - d_{j-1})\mathrm{ReLU}(\mathbf{w}^\top x - t_{j-1})x \tag{31}
$$

A similar expression holds if $\mathbf{w}^\top x$ is negative. This expression is asymptotically quadratic in $\|x\|$.

To verify Assumptions 3 and 6 in Proposition 1, we need only consider the derivatives (30) and (31). We have that $\|\nabla_\theta g_\theta(x)\|$ is asymptotically bounded by a quadratic in $\|x\|$, as required for Fisher's identity given finite second moment. Moreover, since the derivatives are sums and products of ReLU and squared ReLU units, Lipschitz continuity of $\nabla_\theta \log p(y|\theta)$ on compact $\Theta$ comes from the dominated convergence theorem.

## C  Time comparison

For TV, we report the time required to reconstruct 50 images using 5000 iterations of gradient descent for a fixed regularization parameter. The parameter is chosen via a manual grid search to maximize PSNR.

For DIP, we compared 4 different training settings (2 architectures and 2 initializations) and reported the best metrics for the Gaussian deconvolution and Poisson denoising experiments. We report the average time required to reconstruct using DIP by dividing the total time to reconstruct 50 images using all 4 methods by 4.

Table 4: Approximate time required to train and test the compared methods on Gaussian deconvolution and Poisson denoising tasks. Testing is done on 50 images, unrolled for 5000 iterations for energy-based methods such as CRR-based and TV.

| | CRR-based | | | | End-to-end/model-based | | | |
| | GS | SAPG | | | EI | TV | DIP | |
| | | MAP | MMSE ($2 \times 10^4$) | MMSE ($1 \times 10^5$) | | | Adam | GD |
|---|---|---|---|---|---|---|---|---|
| Train (Gaussian) | 10 mins | | 3 days | | 4 hours | - | 50 mins | 4 days |
| Train (Poisson) | 45 mins | | 1 week | | 1 day | - | 45 mins | 21 hours |
| Test | 5 secs | 3 mins | 11 mins | 52 mins | 5 secs | 8 secs | - | - |

