# OpenReview forum: "Unsupervised Training of Convex Regularizers using Maximum Likelihood Estimation"
_TMLR — Accepted by TMLR_

### Review · Reviewer_HnW1 · 2024-09-23

**Summary Of Contributions:**

This work considers solving inverse problems with learned regularization in situations where one does not have access to samples of ground-truth images and only access to noisy measurements. The authors consider parameterizing a regularizer via a neural network-based convex ridge regularizer (CRR) $g_{\theta}$ and use an empirical Bayesian approach to maximize $p(y | \theta)$, where $y$ constitute our measurements. To overcome the intractability of computing $\nabla_{\theta} p(y|\theta)$, the authors show that gradient ascent on $p(y|\theta)$ amounts to computing expectations of the gradient of $g_{\theta}$ over the prior and posterior. To compute empirical averages of such expectations, an additional MCMC loop is applied. Once trained, the regularizer can then be used to compute an MAP or MMSE estimate for downstream inverse problems. Theoretically, the authors show that, when using a CRR, the bias of their sampling approach is asymptotically bounded by the step size parameters. They experimentally verify their approach on Gaussian deconvolution and Poisson denoising, showing that they obtain solid performance when learning only on noisy measurement data.

**Audience:**

Yes

**Claims And Evidence:**

Yes

**Requested Changes:**

**Requested changes:**
Overall, I am positive on the paper and think that this is a solid submission to TMLR. The only requested change I have would be to include further discussions on limitations (e.g., runtime of the training algorithm and focusing on convexity), as this was not thoroughly discussed in the main paper or in the appendix.

**Comments/questions:**
- One of the main questions I have is in regards to runtime and how long it takes to train such a regularizer. How does this compare to, e.g., the adversarial regularization framework of Lunz et al? A more in-depth discussion about this would be nice and improve the quality of the paper.
- Under the assumptions on the CRR, you effectively have a strongly log-concave density. Can the authors comment on whether one can obtain stronger sampling guarantees (e.g., non-asymptotic ones) with this additional structure?
- One should note the structure of the noisy images $x_i^{\mathrm{noisy}}$ in equation (18). Are these simply backprojected images, as in Lunz et al, or are these clean images contaminated with noise?
- It is noted that DIP has a lack of theoretical guarantees. While I agree in general it is true one cannot achieve convergence, it could be good to reference a few papers in the intro that have addressed this in simpler settings (using NTK-based techniques), e.g.,

"Denoising and Regularization via Exploiting the Structural Bias of Convolutional Generators", R. Heckel & M. Soltanolkotabi, ICLR 2020


**Typos:**
- Should be a parentheses around $E_{\theta} + \iota_{+}$.
- The notation for Borel sets $\mathcal{B}(\mathbb{R}^d)$ should be defined.
- In assumption 5, does the notation refer to the complement of the ball $B(0,R_1)$? I would potentially choose a different notation for the constant $c > 0$.
- Top of page 12: the weight matrix is not bolded.
- There is an exclamation mark in equations describing the Poisson denoising problem.

**Strengths And Weaknesses:**

**Strengths:**
- The work tackles a timely and important problem, namely addressing how to learn data-driven regularizers in situations when one only has access to noisy measurements.
- The paper is well-written and easy to read, while also providing a nice background on learned variational regularization and its connection to bayesian methods.
- Under the assumptions of parameterizing $g_{\theta}$ as a CRR, the authors can obtain nice theoretical guarantees on the asymptotic convergence in expectation of their sampling procedure. The result is mostly an application of previous work, but some work is necessary to show that the CRR indeed satisfies the assumptions necessary for the Theorem.
- The performance is not far behind comparable supervised methods, with nice unforeseen properties such as stronger generalization properties with respect to the forward operator.

**Weaknesses:**
- Through the use of two MCMC methods to optimize $p(y |\theta)$, this method can be potentially very slow. The theoretical results are also asymptotic in nature, which do not necessarily give guarantees in terms of how long it would take to obtain a high-quality sample. Based on my reading, there is no discussion on how long it takes to train the regularizer, which I think is important.
- Focusing on convexity, while certainly nice in terms of the induced posterior's theoretical properties, does come with insurmountable limitations. In particular, in ill-posed problems where the posterior can be multi-modal (e.g., when the forward model is nonlinear), such convex regularizers can never capture this structure since log-concave distributions are necessarily uni-modal.

---

> ### Author Response · Authors · 2024-11-06
> **Response to Reviewer HnW1**
>
> We thank the reviewer for the prompt review and thoughtful comments. Please find below point-by-point responses to the concerns raised.
>
> > Through the use of two MCMC methods to optimize $p(y|\theta)$, this method can be potentially very slow. The theoretical results are also asymptotic in nature, which do not necessarily give guarantees in terms of how long it would take to obtain a high-quality sample. Based on my reading, there is no discussion on how long it takes to train the regularizer, which I think is important.
>
> Compared to non-sampling based approaches such as EI, the SAPG method takes longer to train as well as test, due to the computational burden of sampling as pointed out by the reviewer. For Gaussian deconvolution, training EI and DIP take around 1 hour, while SAPG takes around 3 days. For Poisson denoising, DIP takes around 1 hour, EI takes around 1 day (due to the severe regularization needed near 0), and SAPG around 1 week. We note that this gap during training is still significantly smaller than the typical gap during inference, where sampling can take thousands of times longer compared to end-to-end networks. Reconstructing 50 samples takes less than a second for end-to-end methods, sampling $10^6$ iterations from SAPG takes around 20 minutes, and computing a MAP estimator from the trained CRR using 5000 optimization iterations takes around 3 minutes.
>
> We note at the end of Section 3 that the inner MCMC loop (step 6 in the algorithms) to approximate the prior/posterior integrals, as noted in the reviewer's summary, can be done with only one sampling step.
>
> > Focusing on convexity, while certainly nice in terms of the induced posterior's theoretical properties, does come with insurmountable limitations. In particular, in ill-posed problems where the posterior can be multi-modal (e.g., when the forward model is nonlinear), such convex regularizers can never capture this structure since log-concave distributions are necessarily uni-modal.
>
> Indeed, convexity plays a central role in our methods by improving their convergence properties and by greatly simplifying their analysis, similarly to optimisation methods. We agree that it would be interesting and valuable to extend our methods to non-convex settings. The analysis of Durmus et al. [2018] that we leverage in this paper can be extended to non-convex settings, and we are particularly interested in the case where the prior $g_\theta$ is (weakly) convex and coercive, rather than strongly convex. This would allow us to engage a recent line of work by Shumaylov et al. that demonstrates better performance of weakly convex regularizers Shumaylov et al. [2023, 2024], where convergence is given using the Kurdyka-Lojasiewicz property. Combining our proposed method with that approach  would allow for more expressiveness, easier training, and theoretically remove the uni-modal restriction.
>
> **Reviewer Comments/Questions**
> > One of the main questions I have is in regards to runtime and how long it takes to train such a regularizer. How does this compare to, e.g., the adversarial regularization framework of Lunz et al? A more in-depth discussion about this would be nice and improve the quality of the paper.
>
> We have added table and comparison of training time and runtimes of the methods used in the paper in the appendix. Towards adversarial regularization, we note that unsupervised learning methods typically take longer to train than supervised methods.
>
> > Under the assumptions on the CRR, you effectively have a strongly log-concave density. Can the authors comment on whether one can obtain stronger sampling guarantees (e.g., non-asymptotic ones) with this additional structure?
>
> We note that it is possible to strengthen the theorem presented in the paper to
> non-asymptotic under a different step-size regime. De Bortoli et al. [2021] provides a detailed
> analysis of the strongly log-concave setting, and experimental verification with a strongly log-
> concave Bayesian logistic regression experiment.
>
> > One should note the structure of the noisy images in equation (18). Are these simply back-projected images, as in Lunz et al, or are these clean images contaminated with noise?
>
> The noisy images when warmstarting are simply the corrupted (blurred + noisy in the case of Gaussian) images, as we use this for initialization. We have added this in the text.
>
> > It is noted that DIP has a lack of theoretical guarantees. While I agree in general it is true one cannot achieve convergence, it could be good to reference a few papers in the intro that have addressed this in simpler settings (using NTK-based techniques), e.g. [Heckel and Soltanolkotabi 2020]
>
> We have added a sentence referring to some theoretical results related to DIP, including the suggested reference.

---

> ### Author Response · Authors · 2024-11-06
> **Response to Reviewer HnW1 (ctd)**
>
> **Typos**
> > Should be a parentheses around $E_\theta + \iota_+$
>
> Added.
>
> > The notation for Borel sets $\mathcal{B}(\mathbb{R}^d)$ should be defined.
>
> We have added a clarification in Assumption 3.
>
> > In assumption 5, does the notation refer to the complement of the ball $B(0,R_1)$? I would potentially choose a different notation for the constant $c>0$.
>
> Yes. We have instead changed the notation for the complement to make the distinction with the constant $\mathrm{c}$ clearer.
>
> > Top of page 12: the weight matrix is not bolded.
>
> Fixed
>
> > There is an exclamation mark in equations describing the Poisson denoising problem.
>
> The exclamation mark is notation for factorial, as the $y_i$ are integer valued. We have added clarification at the relevant locations.
>
> **References**
>
> Valentin De Bortoli, Alain Durmus, Marcelo Pereyra, and Ana F Vidal. Efficient stochastic optimisation by unadjusted Langevin Monte Carlo: Application to maximum marginal likelihood and
> empirical Bayesian estimation. Statistics and Computing, 31:1–18, 2021.
>
> Alain Durmus, Eric Moulines, and Marcelo Pereyra. Efficient Bayesian computation by proximal
> Markov chain Monte Carlo: when Langevin meets Moreau. SIAM Journal on Imaging Sciences,
> 11(1):473–506, 2018.
>
> Zakhar Shumaylov, Jeremy Budd, Subhadip Mukherjee, and Carola-Bibiane Sch¨onlieb. Provably
> convergent data-driven convex-nonconvex regularization. In NeurIPS 2023 Workshop on Deep
> Learning and Inverse Problems, 2023.
>
> Zakhar Shumaylov, Jeremy Budd, Subhadip Mukherjee, and Carola-Bibiane Sch¨onlieb. Weakly convex regularisers for inverse problems: Convergence of critical points and primal-dual optimisation.
> arXiv preprint arXiv:2402.01052, 2024.

---

### Review · Reviewer_bTr2 · 2024-10-21

**Summary Of Contributions:**

This paper addresses the inverse problem of reconstructing images from noisy measurements. By leveraging maximum likelihood estimation (MLE), the authors propose an unsupervised method to train convex neural network-based image regularizers using only noisy measurements without requiring ground-truth clean data. This way, they attempt to extend existing unsupervised methods with a more expressive model for the regularizer. In terms of efficiency and scalability, the paper introduces a scalable method that extends the stochastic approximation proximal gradient (SAPG) algorithm that handles large datasets via mini-batching and efficient Markov Chain Monte Carlo (MCMC) sampling to overcome computational challenges. The authors also provide theoretical convergence guarantees for their method. The paper provides proof of ergodic convergence for the proposed SAPG algorithm using a convex ridge regularizer architecture. Their experimental results show that the proposed method performs comparably to state-of-the-art supervised approaches across two image corruption tasks, Gaussian deconvolution and Poisson denoising. They also showed better generalization compared to end-to-end supervised methods.

**Audience:**

Yes

**Claims And Evidence:**

Yes

**Requested Changes:**

Critical:

- Incorporating additional evaluation metrics
- Including recent state-of-the-art methods for comparison (diffusion-based models)
- Using Vidal et al. as a baseline
- Clarifying the novelty in convergence proof
- Clarifying novelty in mini-batching approach

Strengthening:

- Expanding the diversity of datasets used in experiments
- Expanding the diversity of inverse problems (e.g., MRI reconstruction, super-resolution)

**Strengths And Weaknesses:**

Strengths:
The paper is well-written and clearly presents the problem statement, relevant background, and literature review. Also, the explanations of the proposed method and theoretical foundations are easy to follow. By incorporating a more general and expressive regularizer architecture, the proposed method extends traditional approaches like total variation, making it adaptive to different applications. The paper addresses various aspects by including a theoretical convergence proof for the proposed method, as well as proposing a scalable and efficient approach that handles large datasets using mini-batching and MCMC techniques.

Weaknesses:
 The experiments are somewhat limited in terms of dataset variety and evaluation metrics. Only PSNR has been considered as an evaluation metric, while other metrics such as SSIM (structural similarity index measure) could provide a more comprehensive assessment of recovered image quality. Also, while DIP (Deep Image Prior) is used as a baseline, it is relatively old compared to recent state-of-the-art methods. Considering more recent methods, especially diffusion-based models, would provide a better comparison for the effectiveness of the proposed approach. For example, the paper misses including recent works such as:

- Feng, Berthy T., et al. (2023): "Score-based diffusion models as principled priors for inverse imaging."
- Chung, Hyungjin, et al. (2023): "Diffusion Posterior Sampling for General Noisy Inverse Problems."
- Zhu, Yuanzhi, et al. (2023): "Denoising diffusion models for plug-and-play image restoration."

Moreover, the method builds upon prior work by Vidal et al., but the original work from Vidal has not been considered as a baseline. It is important to compare against this to show the improvements clearly. The experiments focus primarily on Gaussian deconvolution and Poisson denoising. Including more diverse inverse problems (such as MRI reconstruction or super-resolution) would be interesting to show the broader applicability of the proposed method. Regarding the convergence proof, there is a question of novelty in this aspect. Similar proofs for convergence exist in the literature for algorithms that involve stochastic gradient descent (SGD) or stochastic optimization methods that combine gradient updates with Markov Chain Monte Carlo (MCMC) techniques. The proof of convergence in this paper also builds on existing work, and the authors specifically cite De Bortoli et al. (2021) as the foundation for their proof, which provides convergence guarantees for Langevin algorithms that use gradient-based methods combined with MCMC for sampling from complex distributions.
Additionally, the method itself is not entirely new, as mini-batching for large datasets is a well-established approach. So, while the paper is well-written, the overall novelty appears to be limited, and further clarification on what sets this work apart from existing methods is required. Moreover, the experiments could be extended by incorporating more diverse datasets, evaluation metrics, and stronger baselines to show the method’s effectiveness better.

---

> ### Author Response · Authors · 2024-11-06
> **Response to Reviewer bTr2**
>
> We thank the reviewer for the comprehensive summary. Please find below the point-by-point responses to the reviewer's concerns.
>
> > The experiments are somewhat limited in terms of dataset variety and evaluation metrics. Only PSNR has been considered as an evaluation metric, while other metrics such as SSIM (structural similarity index measure) could provide a more comprehensive assessment of recovered image quality.
>
> We have added SSIM as an additional quality metric. We leave alternative imaging experiments such as MRI reconstruction to future work, due to computational and time constraints.
>
> > While DIP (Deep Image Prior) is used as a baseline, it is relatively old compared to recent state-of-the-art methods. Considering more recent methods, especially diffusion-based models, would provide a better comparison for the effectiveness of the proposed approach.
>
> We choose DIP since it is unsupervised, in that no ground truth image is used (other than choosing the stopping time), no pre-trained model is used either, and no multiple copies of noisy data are given. We note that there are recent unsupervised diffusion model works based on Stein's unbiased risk estimator (SURE) [1,2], but these works assume the existence of multiple noisy copies of corrupted data (like Noise2Self, Noise2Noise). The authors are not aware of any diffusion model works that satisfy the notion of unsupervised learning in our work, specifically of single-shot corrupted datasets.
>
> > Moreover, the method builds upon prior work by Vidal et al., but the original work from Vidal has not been considered as a baseline. It is important to compare against this to show the improvements clearly.
>
> The original work by Vidal et al. requires a hand-crafted prior, which was taken to be TV or total generalized variation, and predicts the regularization parameters. Therefore, the method performance is determined by the best possible regularization of the given prior, which we alternatively compute using a simpler grid search. We have added a small explanation in the experiments section that the reported TV value will be higher than if Vidal et al. is used, due to the regularization parameter grid search.
>
> > The experiments focus primarily on Gaussian deconvolution and Poisson denoising. Including more diverse inverse problems (such as MRI reconstruction or super-resolution) would be interesting to show the broader applicability of the proposed method.
>
> We leave this to future work due to computational limits and tight time constraints of the review cycle. We expect that the proposed method should still work where the convex ridge regularizer is effective.
>
> > Regarding the convergence proof, there is a question of novelty in this aspect. Similar proofs for convergence exist in the literature for algorithms that involve stochastic gradient descent (SGD) or stochastic optimization methods that combine gradient updates with Markov Chain Monte Carlo (MCMC) techniques. The proof of convergence in this paper also builds on existing work, and the authors specifically cite De Bortoli et al. (2021) as the foundation for their proof, which provides convergence guarantees for Langevin algorithms that use gradient-based methods combined with MCMC for sampling from complex distributions.
>
> As noted correctly by the reviewer, the theoretical contributions in this work consist of verifying the assumptions of De Bortoli et al., which addresses the problem of convergent training. The convex architecture allows for convergence of both the stochastic optimization process, as well as the sampling/optimization process for MMSE/MAP estimation during inference.

---

> > ### Author Response · Authors · 2024-11-06
> > **Response to Reviewer bTr2 (ctd)**
> >
> > > Additionally, the method itself is not entirely new, as mini-batching for large datasets is a well-established approach. The overall novelty appears to be limited, and further clarification on what sets this work apart from existing methods is required.
> >
> > Mini-batching is introduced as a computational tool to alleviate memory problems from working with large data. This is necessary as we extend previous work from single images to image datasets.
> >
> > The novelty of the proposed method comes from a combination of the problem setting and methodology, as stated in the introduction:
> >
> > * We demonstrate that our method works in the difficult unsupervised setting of a one-shot corrupted dataset (only single copies of noisy images). Most other unsupervised methods (excluding equivariant imaging) consider the setting where multiple copies of noisy images are available.
> > * Related to reconstruction from a single corrupted image (Vidal et al.), we extend both the dataset and model expressivity, turning training into a high-dimensional Bayesian optimization problem. This necessitates computational tools such as mini-batching.
> > * We demonstrate that the gap between unsupervised and supervised learning for the same architecture is small. We do this using the comparison with the gradient-step training for convex ridge regularizers.
> >
> > **References**
> >
> > [1] Aali, Asad, et al. "Solving inverse problems with score-based generative priors learned from noisy data." 2023 57th Asilomar Conference on Signals, Systems, and Computers. IEEE, 2023.
> >
> > [2] Kawar, B., Elata, N., Michaeli, T., \& Elad, M. GSURE-Based Diffusion Model Training with Corrupted Data. Transactions on Machine Learning Research.

---

> > > ### Comment · Reviewer_bTr2 · 2024-11-25
> > > **Limitation in novelty and comparative evaluation of diffusion-based approaches to image inverse problems**
> > >
> > > I appreciate the author's discussion of diffusion-based unsupervised approaches to image inverse problems. While the proposed methodology differed from these approaches in that it does not rely on any pre-trained prior image model and instead learns some parametric prior directly from the observation, it would still be relevant and interesting to include those diffusion-based methods in the experiments to have a better view of their comparative performance and possible disadvantages. Currently, I still find the experiments rather limited to truly evaluating the significance of the proposed method and properly positioning it with respect to the wide range of supervised, unsupervised, and self-supervised approaches to inverse problems.
> > >
> > > Also, regarding the novelty of the work, I find the explanation provided quite general. Although it is acknowledged that the proposed method builds on existing techniques and adapts them to a new problem setting, still, the outlined contributions, such as the use of mini-batching and the adaptation to a one-shot corrupted dataset, heavily rely on prior works.
> > >
> > > So, given the limited experiments and my concerns regarding the novelty and contribution of the paper, I believe this paper requires a major revision.

---

### Review · Reviewer_trdP · 2024-10-24

**Summary Of Contributions:**

The papers extends previous work for unsupervised learning of convex regularizers for inverse problems. In my view. the main claims are as follows:
1. *Performance*: it performs better than the closest prior work, and almost as well as supervised baselines.
2. *Scalability*: it can be trained in a higher-dimensional setting than the closest prior work.
3. *Theoretically convergent*: given a particular choice of parametrization, the authors verify that the convergence conditions in previous works apply.

**Audience:**

Yes

**Broader Impact Concerns:**

None.

**Claims And Evidence:**

No

**Requested Changes:**

- Relate the work to diffusion models
- Add diffusion-based baselines to the experiments
- Provide additional evidence for the claims, as suggested above.

**Strengths And Weaknesses:**

**Strengths**

- *Clear presentation*: the writing is clear; the motivation and literature is thoroughly grounded in the inverse problems literature.
- *Theoretical results*: I did not check the proofs in detail, but their statements and conditions are clear.
- *Aims*: expressive parameterizations seems to be key to reach state-of-the-art performance, hence the associated improvements to scalability are appreciated.

**Weaknesses**

*Major weaknesses*

I see two related weaknesses due to which I don't think the current submission is "supported by accurate, convincing and clear evidence".
1. *Diffusion models are notably absent.*
    It is by now well-established that diffusion models can provide strong priors for inverse problems [1-7], but there is no discussion or comparison with this class of methods. First, this makes it hard to position the paper. Second, I believe that many state-of-the-art models for inverse problems are in fact based on diffusions, hence the lack of such baselines significantly weakens the performance claims of the proposed method. Most relevant would be unsupervised baselines such as [8] and [9], but to support the comparison to supervised methods in claim 1 there are also many supervised methods like [10].

2. *The experiments are only weakly aligned with claims.*

    - Regarding claim 1 above, I might have misunderstood this, but I don't think the original work is included as a baseline, which would the obvious way to provide evidence for the claim.
    - Regarding claim 2, there is no experimental evidence supporting the claims of improved scalability due to the batched training.
    - Regarding claim 3, the authors acknowledge that batched SAPG-ULA is biased. Since this bias isn't analyzed theoretically, I would have liked to see an experimental evaluation of its size and effect.
    - I think the experiment in section 4.3 is of some interest, but it has no bearing on the claims and, besides, would probably merit a different approach.

*Minor weaknesses*

- Only using PSNR is a bit meagre. Computer vision papers usually include also a perceptual metric like FID or LPIPS. In addition, I think it could make sense to compare the marginal likelihood as well in this case.
- I think the paper would benefit from being shortened. Some suggestions:
    - shorten each paragraph of the literature review, but add one on diffusion models
    - move sections 2.4 and 3.1 to appendix

*Questions:*
1. Would it be possible to generalize the class of regularizers to Input convex neural networks [11]?
2. Would it be possible to use variance reduction techniques like common random numbers?

*References*

[1] Feng et al., Score-Based Diffusion Models as Principled Priors for Inverse Imaging. ICCV, 2023.

[2] Graikos et al., Diffusion Models as Plug-and-Play Priors. NeurIPS, 2022.

[3] Cardoso et al., Monte Carlo guided Diffusion for Bayesian linear inverse problems. ICLR, 2024

[4] Mardani et al., A Variational Perspective on Solving Inverse Problems with Diffusion Models. ICLR, 2024.

[5] Dou and Song, Diffusion posterior sampling for linear inverse problem solving: A filtering perspective. ICLR, 2024.

[6] Luo et al., "Taming diffusion models for image restoration: A review." arXiv preprint 2409.10353, 2024.

[7] Zhao et al. Conditional sampling within generative diffusion models. arXiv preprint 2409.09650, 2024.

[8] Kawar et al., Denoising diffusion restoration models. NeurIPS, 2022.

[9] Wang et al., Zero-shot image restoration using denoising diffusion null-space model. ICLR, 2023.

[10] Luo et al., 2023. Image restoration with mean-reverting stochastic differential equations. ICML, 2023.

[11] Amos et al., Input convex neural networks. ICML, 2017.

---

> ### Author Response · Authors · 2024-11-06
> **Response to Reviewer trdP**
>
> We thank the reviewer for the detailed comments and references. Please find point-by-point responses to the reviewer's concerns below.
>
> > Diffusion models are notably absent. It is by now well-established that diffusion models can provide strong priors for inverse problems [1-7], but there is no discussion or comparison with this class of methods. First, this makes it hard to position the paper. Second, I believe that many state-of-the-art models for inverse problems are in fact based on diffusions, hence the lack of such baselines significantly weakens the performance claims of the proposed method. Most relevant would be unsupervised baselines such as [8] and [9], but to support the comparison to supervised methods in claim 1 there are also many supervised methods like [10]. (Add diffusion-based baselines to the experiments/Provide additional evidence for the claims, as suggested above.)
>
> In our work and more generally across inverse problems, an unsupervised method does not allow access to ground truth during training. As far as the authors are aware, unsupervised diffusion models that can be trained using only single corrupted measurements have not been explored in the literature (though references exist for multiple noisy measurements, based on Stein's unbiased risk estimator). After checking references [8,9], their notion of unsupervised learning is different from the sense of our work, as they both use pre-trained diffusion models. Training a diffusion model without ground truth data and only single copies of noisy data is an interesting gap in the literature, which might be doable with similar Bayesian tools used in our work.
>
> For additional experiments comparing with diffusion models as priors, we believe that the state-of-the-art supervised diffusion model paradigm is too far from our setting for a fair comparison. Notably, the gaps come from unsupervised/supervised training, parameter count, and convergence guarantees.
>
> >The experiments are only weakly aligned with claims.
> > Regarding claim 1 above, I might have misunderstood this, but I don't think the original work is included as a baseline, which would the obvious way to provide evidence for the claim.
>
> Previous work considers total variation as the prior. As we use a grid search to find the best possible parameter for TV, the previous work is at most as good as the reported number for TV. While it is possible to change the regularizer to another hand-crafted one, it is not a-priori clear which one to use, which again motivates the use of a learned regularizer. We have clarified this in the manuscript in the experiment exposition.
>
> > Regarding claim 2, there is no experimental evidence supporting the claims of improved scalability due to the batched training.
>
> Computing the likelihoods of a CRR over an entire dataset is computationally infeasible using previous methods. Batching is necessary for training on image datasets due to the significantly increased parameter and data counts, and is introduced as a computational tool to combat memory problems. Previous work only works on reconstructing single images, and we extend this to the typical machine learning paradigm of having a training image dataset.
>
> > Regarding claim 3, the authors acknowledge that batched SAPG-ULA is biased. Since this bias isn't analyzed theoretically, I would have liked to see an experimental evaluation of its size and effect.
>
> We refer to De Bortoli et al. and Vidal et al. for experimental verification of bias in cases where the ground truth parameter maximum marginal likelihood estimator is known (while it is not in our case). In particular, Section 4 of De Bortoli et al. considers the bias for Bayesian logistic regression and compression, and Vidal et al. consider bias for various low-dimensional examples. The papers find that the bias is not significant if the hyperparameter guidelines are follows, as we do in our work.
>
> > I think the experiment in section 4.3 is of some interest, but it has no bearing on the claims and, besides, would probably merit a different approach.
>
> We include the change in forward operator to provide some empirical justification to the choice of learned regularizer as opposed to end-to-end. Within the same CRR architecture, we also observe slightly better generalization performance, which could be of independent interest. We leave more sophisticated approaches to constructing more transferable architectures for EI for future work.

---

> > ### Author Response · Authors · 2024-11-06
> > **Response to Reviewer trdP (ctd)**
> >
> > > Only using PSNR is a bit meagre. Computer vision papers usually include also a perceptual metric like FID or LPIPS. In addition, I think it could make sense to compare the marginal likelihood as well in this case.
> >
> > We have added SSIM as a standard metric in related imaging literature. As FID and LPIPS appear to be tailored more towards generative models, we believe PSNR and SSIM to be more representative of the reconstruction quality. Could the reviewer please clarify their meaning of marginal likelihood, as our definition $p(y|\theta)$ is intractable.
> >
> > > I think the paper would benefit from being shortened. Some suggestions: shorten each paragraph of the literature review, but add one on diffusion models; move sections 2.4 and 3.1 to appendix
> >
> > We have added a short paragraph on diffusion models and the varying notions of unsupervised learning in this area, and moved Section 3.1 to the appendix. We keep Section 2.4 as it is needed to explain the adversarial warm-starting with small amounts of ground truth.
> >
> > **Reviewer Questions**
> > > Would it be possible to generalize the class of regularizers to Input convex neural networks [11]?
> >
> > The theory should still hold for ICNNs. Goujon, Neumayer, and Unser (2024) claim that in the learned regularizer setting, the convex ridge regularizer architecture performs on par with ICNNs with significantly fewer (order of $10^4$ instead of $10^7$) parameters.
> >
> > > Would it be possible to use variance reduction techniques like common random numbers?
> >
> > We refer the reviewer to Haji-Ali et al., which considers such variance reduction in the diffusion setting. The reference considers score-based diffusion using multiple (multi-level) Monte Carlo instances with coupled noise. While the SDE considered in the, application to ULA can also be done for instance with multiple Markov chains of varying step-sizes, with larger step-sizes providing robust fast approximate convergence, and smaller step-sizes giving details with smaller bias.
> >
> > **References**
> >
> > Goujon, A., Neumayer, S., \& Unser, M. (2024). Learning weakly convex regularizers for convergent image-reconstruction algorithms. SIAM Journal on Imaging Sciences, 17(1), 91-115.
> >
> > Haji-Ali, A. L., Pereyra, M., Shaw, L., \& Zygalakis, K. (2024). Bayesian computation with generative diffusion models by Multilevel Monte Carlo. arXiv preprint arXiv:2409.15511.
> >
> > De Bortoli, V., Durmus, A., Pereyra, M., \& Vidal, A. F. (2021). Efficient stochastic optimisation by unadjusted Langevin Monte Carlo: Application to maximum marginal likelihood and empirical Bayesian estimation. Statistics and Computing, 31, 1-18.
> >
> > Vidal, A. F., De Bortoli, V., Pereyra, M., \& Durmus, A. (2020). Maximum likelihood estimation of regularization parameters in high-dimensional inverse problems: An empirical bayesian approach part i: Methodology and experiments. SIAM Journal on Imaging Sciences, 13(4), 1945-1989.

---

### Decision · Action_Editor_pVSR · 2024-12-06

**Recommendation:** Accept with minor revision

**Comment:**

The paper considers inverse imaging problems, where ground-truth data is unavailable, and only noisy measurements are provided. The authors propose an unsupervised learning approach, where they train a convex neural network-based image regularizer, using maximum marginal likelihood estimation. They extend the stochastic approximation proximal gradient (SAPG) algorithm of (Vidal et al., 2020), from single measurements to datasets, leveraging mini-batching and efficient Markov Chain Monte Carlo (MCMC) sampling to overcome computational challenges.
Theoretical convergence results of the proposed method are provided when using the convex ridge regularizer (CRR).  Experiments demonstrate that their method achieves performance comparable to analogous supervised models for image reconstruction tasks such as Gaussian deconvolution and Poisson denoising, while offering better generalization than end-to-end models.

Below is a summary of the main strengths and weaknesses highlighted by reviewers and based on my own assessment.

Strengths:
- The problem considered is important; namely learning data-driven regularizers from only noisy measurements
- The paper is well written and clear. It provides a clear motivation, background, and explanation of the method and its theoretical results.
- Nice theoretical guarantees on the asymptotic convergence when using CRR. The result follows from prior work, but requires some work to verify that the assumptions necessary hold.
- Proposed method improves over existing MLE-based methods in both model expressiveness and dataset size
- Experimental results demonstrate competitive performance with supervised methods while showing better generalization.

Weaknesses:
- Proposed method is slow. No theoretical bound on the time required to obtain good sample.
- Focus on convex regularizers is limiting. Though it does allow for nice theoretical properties.

The official recommendations from the reviewers were mixed, with two reviewers recommending to reject and one to accept. The authors have addressed in their responses most of the reviewers' concerns. A key concern raised by two reviewers is the lack of comparison with diffusion-based methods. The authors added a discussion noting that existing diffusion models require either clean data or multiple noisy measurements, which is different from the setting here where only single copies of noisy data are available. Given this difference in setting, I agree with Reviewer HnW1 that a comparison with diffusion models, while interesting, is not necessary. The other concern raised by Reviewer bTr2 is about the novelty of the work, which is not a necessary criteria for acceptance at TMLR.  Hence, I am recommending to accept this paper with minor revision.

Requested revisions:
- In the paragraph added on p.10 about empirical studies of bias of SAPG, add the conclusion from these studies, i.e., this comment from your response: "The papers find that the bias is not significant if the hyperparameter guidelines are followed, as we do in our work."
- Add the motivation for the experiments in Section 4.3 mentioned in the response to Reviewer trdP
- Add a discussion of the limitations of the proposed method (runtime, convexity).

**Audience:**

yes

**Claims And Evidence:**

yes